# MITIGATING BIAS IN CALIBRATION ERROR ESTIMATION

## ABSTRACT

Building reliable machine learning systems requires that we correctly understand their level of confidence. Calibration focuses on measuring the degree of accuracy in a model's confidence and most research in calibration focuses on techniques to improve an empirical estimate of calibration error, $ECE_{BIN}$. Using simulation, we show that $ECE_{BIN}$ can systematically underestimate or overestimate the true calibration error depending on the nature of model miscalibration, the size of the evaluation data set, and the number of bins. Critically, $ECE_{BIN}$ is more strongly biased for perfectly calibrated models. We propose a simple alternative calibration error metric, $ECE_{SWEEP}$, in which the number of bins is chosen to be as large as possible while preserving monotonicity in the calibration function. Evaluating our measure on distributions fit to neural network confidence scores on CIFAR-10, CIFAR-100, and ImageNet, we show that $ECE_{SWEEP}$ produces a less biased estimator of calibration error and therefore should be used by any researcher wishing to evaluate the calibration of models trained on similar datasets.

## 1 INTRODUCTION

Machine learning models are increasingly deployed in high-stakes settings like self-driving cars (Caesar et al., 2020; Geiger et al., 2013; Sun et al., 2020) and medical diagnoses (Esteva et al., 2017; 2019; Gulshan et al., 2016) where a model's ability to recognize when it is likely to be incorrect is critical. Unfortunately, such models often fail in unexpected and poorly understood ways, hindering our ability to interpret and trust such systems (Azulay & Weiss, 2018; Biggio & Roli, 2018; Hendrycks & Dietterich, 2019; Recht et al., 2019; Szegedy et al., 2013). To address these issues, calibration is used to ensure that a machine learning model produces confidence scores that reflect the model's ground truth likelihood of being correct (Platt et al., 1999; Zadrozny & Elkan, 2001; 2002).

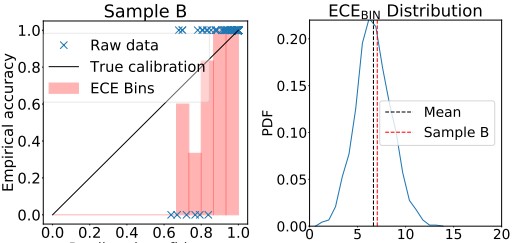

Figure 1: **Bias in $ECE_{BIN}$ for perfectly calibrated models.** Simulated data from a perfectly calibrated model with confidence scores modeled to ResNet-110 CIFAR-10 output (He et al., 2016; Kängsepp, 2019). We show a reliability diagram for a sample of size $n = 200$ and the distribution of $ECE_{BIN}$ scores computed across $10^6$ independent simulations. Even though the model is perfectly calibrated, $ECE_{BIN}$ systematically predicts large calibration errors.

To obtain an estimate of the calibration error, or $ECE^1$, the standard procedure (Guo et al., 2017; Naeini et al., 2015) partitions the model confidence scores into bins and compares the model's predicted accuracy to its empirical accuracy within each bin. We refer to this specific metric as $ECE_{BIN}$. Although recent work has pointed out that $ECE_{BIN}$ is sensitive to implementation hyperparameters (Kumar et al., 2019; Nixon et al., 2019), measuring the statistical bias in $ECE_{BIN}$, or the difference between the expected $ECE_{BIN}$ and the true calibration error (TCE), has remained largely unaddressed.

In this paper, we address this problem by developing techniques to measure bias in existing calibration metrics. We use simulation to create a setting where the TCE can be computed analytically and thus the bias can be estimated directly. As Figure 1 highlights, we find empirically that $ECE_{BIN}$ has non-negligible statistical bias and systematically predicts large errors for perfectly calibrated models.

---

[1]Naeini et al. (2015) introduce ECE as an acronym for *Expected* Calibration Error. However, ECE is not a proper expectation whereas the true calibration error is computed under an expectation. To resolve this confusion, we prefer to read ECE as *Estimated* Calibration Error.

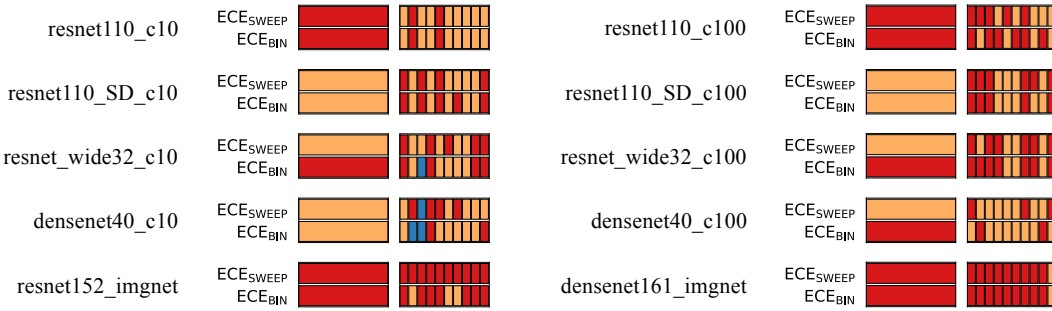

Figure 2: **Bias affects which recalibration algorithm is preferred.** For ten models, we report which recalibration method is determined to be superior based either on ECE$_{\text{BIN}}$ or ECE$_{\text{SWEEP}}$. The wide bar indicates the superior method using entire validation set (mean of X instances); narrow bars each use a random sample of 10% of the original validation set. Recalibration methods tested are histogram binning, temperature scaling, and isotonic regression.

Motivated by monotonicity in true calibration curves arising from trained models, we develop a simple alternate for measuring calibration error, the *monotonic sweep calibration error* (ECE$_{\text{SWEEP}}$), which chooses the largest number possible while maintaining monotonicity in the approximation to the calibration curve. Our results suggest that ECE$_{\text{SWEEP}}$ is less biased than the standard ECE$_{\text{BIN}}$ and can thus more reliably estimate calibration error.

Does the use of an improved ECE measure affect which recalibration method is preferred? In Figure 2, we examine this question using 10 pre-trained models, and compare the standard ECE measure, ECE$_{\text{BIN}}$ with 15 equal-width-spaced bins, to our ECE$_{\text{SWEEP}}$. With large dataset sizes for recalibration and evaluation [2], we find that ECE$_{\text{BIN}}$ produces a different selection of the preferred recalibration method on 30% of the models. (We use histogram binning (Zadrozny & Elkan, 2001), temperature scaling (Guo et al., 2017), and isotonic regression (Zadrozny & Elkan, 2002) as the recalibration techniques.) When we reduce the size of the validation and evaluation by 10% and recalibrate with these smaller sets, ECE$_{\text{BIN}}$ produces a different selection on 22% of the cases ( with 10 bins, we see disagreement on 27% of the cases). Thus, the use of our improved ECE measure has significant implications not only for estimation of calibration error but for improving calibration with methods like temperature scaling.

## 2 BACKGROUND

Consider a binary classification setup with input $X \in \mathcal{X}$, target output $Y = \{0, 1\}$, and we have a model $f : X \to [0, 1]$ whose output represents a confidence score that the true label $Y$ is 1.

**True calibration error (TCE).** We define true calibration error as the difference between a model's predicted confidence and the true likelihood of being correct under the $\ell_p$ norm:

$$\text{TCE}(f) = \left( \mathbb{E}_X \left[ |f(X) - \mathbb{E}_Y[Y|f(X)]|^p \right] \right)^{\frac{1}{p}}. \quad (1)$$

The TCE is dictated by two independent features of a model: (1) the distribution of confidence scores $f(x) \sim \mathcal{F}$ over which the outer expectation is computed, and (2) the true calibration curve $\mathbb{E}_Y[Y | f(X)]$, which governs the relationship between the confidence score $f(x)$ and the empirical accuracy (see Figure 3 for illustration).

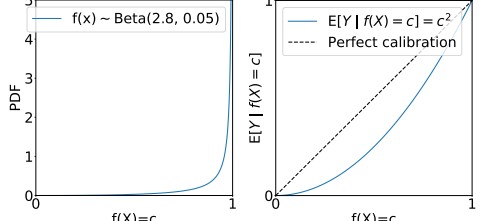

Figure 3: **Curves controlling true calibration error.** Our ability to measure calibration is contingent on both the confidence score distribution (e.g., $f(X) \sim \text{Beta}(2.8, 0.05)$) and the true calibration curve (e.g., $\mathbb{E}_Y[Y | f(X) = c] = c^2$.

In our experiments, we measure calibration error using the $\ell_2$ norm because it increases the sensitivity of the error metric to extremely poorly calibrated predictions, which tend to be more harmful in applications. In addition, the mean squared prediction error of the classifier, or Brier score (Brier, 1950), can be decomposed into terms corresponding to the squared $\ell_2$ calibration error and the variance of the model's correctness likelihood (Kuleshov & Liang, 2015; Kumar et al., 2019).

---

[2] We use standard validation sets of size $5,000$ examples for CIFAR-10/100 and $25,000$ examples for ImageNet and evaluation sets of size $10,000$ for CIFAR-10/100 and $25,000$ for ImageNet.

## 2.1 Estimates of calibration error

To estimate the TCE of a model $f$, assume we are given a dataset containing $n$ samples, $\{x_i, y_i\}_{i=1}^n$. We can approximate TCE by replacing the outer expectation in Equation 1 by the sample average and replacing the inner expectation with an average over a set of instances with similar $f(x)$ values:

$$\text{ECE}_{\text{NEIGH}}(f) = \left( \frac{1}{n} \sum_{i=1}^n \left| f(x_i) - \frac{1}{|\mathcal{N}_i|} \sum_{j \in \mathcal{N}_i} y_j \right|^p \right)^{\frac{1}{p}}, \tag{2}$$

where $\mathcal{N}_i$ is instance $i$'s set of neighbors in model confidence output space.

**Label-binned calibration error (ECE$_{\text{LB}}$).** The label-binned calibration error uses binning to define $\mathcal{N}_i$ and estimate the model's empirical accuracy $\mathbb{E}[Y|f(X)]$. Specifically, the instances are partitioned into $b$ bins, where $B_k$ denotes the set of all instances in bin $k$, allowing us to express Equation 2 in terms of the binned neighborhood:

$$\text{ECE}_{\text{LB}}(f) = \left( \frac{1}{n} \sum_{k=1}^b \sum_{i \in B_k} |f(x_i) - \bar{y}_k|^p \right)^{\frac{1}{p}}, \text{ where } \bar{y}_k = \frac{1}{|B_k|} \sum_{j \in B_k} y_j. \tag{3}$$

Binning is commonly implemented using either *equal width binning* (Guo et al., 2017; Naeini et al., 2015), which creates bins by dividing the model confidence domain $[0, 1]$ into equal sized intervals, or *equal mass binning* (Nixon et al., 2019), which creates bins by dividing the $n$ samples into partitions with an equal number of instances.

**Binned calibration error (ECE$_{\text{BIN}}$).** In contrast to ECE$_{\text{LB}}$, which operates on the original instances but uses binning to estimate empirical accuracy, ECE$_{\text{BIN}}$ collapses all instances in a bin into a single instance and compares the per-bin empirical accuracy to the per-bin confidence score, weighted by the per-bin number of instances. Given $b$ bins, where $B_k$ is the set of instances in bin $k$, and letting $\bar{f}_k$ and $\bar{y}_k$ be the per-bin average confidence score and label, ECE$_{\text{BIN}}$ is defined under the $\ell_p$ norm as

$$\text{ECE}_{\text{BIN}}(f) = \left( \sum_{k=1}^b \frac{|B_k|}{n} \left| \bar{f}_k - \bar{y}_k \right|^p \right)^{\frac{1}{p}} \tag{4}$$

Importantly, ECE$_{\text{BIN}}$ always underestimates ECE$_{\text{LB}}$, $\text{ECE}_{\text{LB}}(f) \geq \text{ECE}_{\text{BIN}}(f)$, which follows by applying Jensen's inequality on each inner term $k \in \{1, 2, \ldots, b\}$ in Eqs. 3 and 4:

$$\frac{1}{|B_k|} \sum_{i \in B_k} |f(X_i) - \bar{Y}_k|^p \geq \left| \sum_{i \in B_k} \bar{f}_k - \bar{Y}_k \right|^p. \tag{5}$$

## 3 Measuring bias through simulation

We focus on bias rather than variance because the variance can be estimated from a finite set of samples through resampling techniques whereas the bias is an unknown quantity that reflects systematic error. We also found empirically that the variance seems relatively insensitive to the estimation technique and number of bins (see Appendix B). The *bias* of a calibration error estimator, ECE$_{\mathcal{A}}$ for some estimation algorithm $\mathcal{A}$, is the difference between the estimator's expected value with respect to the data distribution and the TCE:

$$\text{Bias}_{\mathcal{A}} = \mathbb{E}[\text{ECE}_{\mathcal{A}}] - \text{TCE}. \tag{6}$$

If we assume a particular confidence score distribution $\mathcal{F}$ and true calibration curve $T(X) = \mathbb{E}_Y[Y \mid f(X) = c]$ (see Figure 3 for examples), we can compute the TCE by numerically evaluating the integral implicit in the outer expected value of Equation 1.

We then compute a sample estimate of the bias as follows. First, we generate $n$ samples $\{f(x_i), y_i\}_{i=1}^n$ such that $f(x_i) \sim \mathcal{F}$ and $\mathbb{E}_Y[Y \mid f(X) = c] := T(c)$, and compute the ECE on the sample. We repeat this process for $m$ simulated datasets and compute the sample estimate of bias as the difference between the average ECE and the TCE:[3]

$$\widehat{\text{Bias}}_{\mathcal{A}}(n) = \frac{1}{m} \sum_{i=1}^m \text{ECE}_{\mathcal{A}} - \text{TCE}. \tag{7}$$

**Bias in ECE$_{\text{BIN}}$ for varying hyperparameters.** Using simulation, we next investigate the bias in ECE$_{\text{BIN}}$ as a function of the number of samples $n$ and the number of bins. We compute ECE$_{\text{BIN}}$ with equal width binning and we assume parametric curves for $f(x)$ and $\mathbb{E}_Y[Y \mid f(X) = c]$ that are fit to the ResNet-110 CIFAR-10 model output (see Section 5.1 for details on how we compute fits). Kumar et al. (2019) Proposition 3.3 shows that any binned version of calibration error systematically underestimates TCE *in the limit of infinite data*.

---

[3]In the remainder of the text, we will use the term "bias" to refer to this sample estimate of bias.

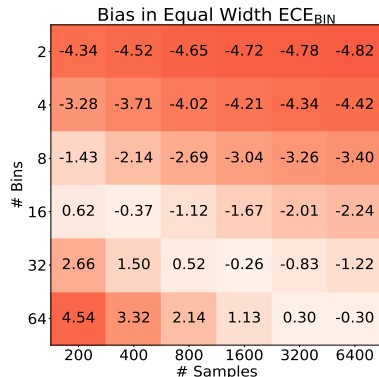

However, for a finite number of samples $n$, Figure 4 shows that $\text{ECE}_{\text{BIN}}$ can either overestimate or underestimate TCE and that increasing the number of bins does not always lead to better estimates of TCE. Intuitively, as the number of bins explodes, each example lies in its own bin, and the prediction error is computed with respect to a target of 0 or 1. Moreover, regardless of binning scheme, there exists a bin number for each sample size that results in the lowest estimation bias and this optimal bin count grows with the sample size. In Appendix B, we show the variance associated with this experiment, and also include results for alternative calibration metrics.

Figure 4: $\text{ECE}_{\text{BIN}}$ with equal width binning can overestimate TCE and the optimal number of bins depends on number of samples.

## 4  MONOTONIC CALIBRATION METRICS

Though Section 3 shows that there exists an optimal number of bins for which $\text{ECE}_{\text{BIN}}$ has the lowest bias, unfortunately, this number depends on the binning technique, the number of samples, the confidence score distribution, and the true calibration curve. This observation motivates us to seek a method for adaptively choosing the number of bins.

Monotonicity in the true calibration curve implies that a model's expected accuracy should always increase as the model's confidence increases. Although such a requirement seems reasonable for most any statistical model, it is not obvious how to prove why or when a "reasonable" model would attain such a property. We offer a rationale for why it should be expected of machine learning models trained with a maximum likelihood objective, e.g., cross-entropy or logistic loss (Murphy, 2012). Namely, from ROC (receiver operating characteristic) analysis of maximum likelihood models, an under-appreciated observation of ROC curves is that a model trained to maximize the likelihood ratio must have a convex ROC curve in the limit of infinite data (Green et al. (1966), Section 2.3). The slope of the ROC curve is related to the calibration curve, and a convex ROC curve implies a monotonically increasing calibration curve (the converse is also true) (Chen et al., 2018; Gneiting & Vogel, 2018).

In practice, several potential confounds may lead to measuring a non-monontonic calibration curve. First, finite data size effects may lead to fluctuations in the true positive or false positive rates, but do not reflect the behavior of the underlying model. Second, deviations in the domain statistics between cross-validated splits in the data may lead to unbounded behavior; however, we assume that such domain shifts are negligible as cross-validated splits are presumed to be selected *i.i.d.*[4] Given that deviations from non-monotonic calibration curves are considered artificial, we posit that any method that is trying to assess the TCE of an underlying model may freely assume monotonicity in the true calibration curve. Note that this proposition already guides the entire field of re-calibration to require that re-calibration methods only consider monotonic functions (Platt et al., 1999; Wu et al., 2012; Zadrozny & Elkan, 2002).

**Monotonic sweep calibration error ($\text{ECE}_{\text{SWEEP}}$).** Accordingly, we leverage the underlying monotonicity in the true calibration and propose the monotonic sweep calibration error, a metric that choose the largest number of bins possible such that it and all smaller bin sizes preserve monotonicity in the bin heights $\bar{y}_k$.

$$\text{ECE}_{\text{SWEEP}} = \max_b \left( \left( \sum_{k=1}^{b'} \frac{|B_k|}{n} \left| \bar{f}_k - \bar{y}_k \right|^p \right)^{\frac{1}{p}} \quad \text{s.t.} \quad \bar{y}_1 \le \bar{y}_2 \ldots \le \bar{y}_{b'}, \ \ \forall b' \le b \right).$$

We can compute the monotonic sweep calibration error by starting with $b = 2$ bins (since $b = 1$ is guaranteed to be a monotonic binning) and gradually increasing the number of bins until we either

---

[4]Note that a third potential reason for a non-monotonic calibration curve is that a classifier could be trained with a non–likelihood-based statistical criteria, e.g. moment matching. However, the lack of monotonic behavior in the calibration curve of such a model may actually be a sign that the model is not reasonable or admissible model for consideration on a given task (Chen et al., 2018; Pesce et al., 2010).

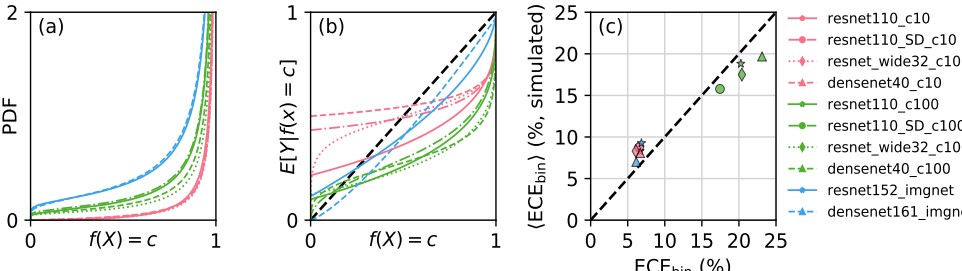

Figure 5: **Maximum likelihood fits to empirical datasets illustrate large skew in their density distribution and calibration function**. For each dataset, (a) confidence distributions were fit with a two-parameter beta distribution and (b) calibration curves were fit via generalized linear models across multiple model families, with the best model selected via the Akaike information criterion (details in Appendix A). Across models, the dataset source systematically affects both curves. (c) We plot the overall quality of the fits by computing the $ECE_{BIN}$ on the original data vs. the $ECE_{BIN}$ averaged over 1000 simulated trials. Curves well-fit to the data should lie close to the identity line.

reach a non-monotonic binning, in which case we return the last $b$ that corresponded to a monotonic binning, or until every sample belongs to its own bin ($b = n$).

---

**Algorithm 1:** Monotonic Sweep Calibration Error

---
1  **for** $b \leftarrow 2$ **to** $n$ **do**
2  $\quad$ Compute $ECE_{BIN}$ bin heights ($\bar{y}_k$) with $b$ bins ;
3  $\quad$ **if** *Binning is not monotonic* **then**
4  $\quad\quad$ b = b -1 ;
5  $\quad\quad$ break ;
6  **return** $ECE_{BIN}$ *computed with* $b$ *bins*

---

## 5 RESULTS

### 5.1 PARAMETRIC FITS CAPTURE CALIBRATION CURVE AND SCORE DISTRIBUTION

TCE is analytically computable when we assume parametric forms for the confidence distribution and the true calibration curve. To what extent can parametric forms capture the diversity and complexity of real world data? In many applications, only sample-based approximations to these functions are available. In order to estimate $ECE_{BIN}$ bias in real-world data, we develop parametric models of empirical logit datasets that enable direct measurement of TCE.

We consider 10 logit datasets (including those studied in Guo et al. (2017)), arising from training four different neural model families (ResNet, ResNet-SD, Wide-ResNet, and DenseNet) on three different image datasets (CIFAR-10/100 and ImageNet) (Deng et al., 2009; He et al., 2016; Huang et al., 2016; 2017; Krizhevsky, 2009; LeCun et al., 1998; Zagoruyko & Komodakis, 2016). For each dataset (Figure 5), we compute confidence scores by applying softmax and top-1 selection to logits from Kängsepp (2019).

Computing TCE directly for real-world data via Equation 1 is infeasible because of the expectation across $X$. Instead, we model the distribution of the scores $f(X)$ directly with a two-parameter beta distribution, which we fit using maximum likelihood estimation (note that in many cases, the confidence scores are heavily skewed). Calibration functions are computed by fitting multiple (binary) generalized linear models (GLM) to the calibration data. From these candidate models, we select the model of best fit using the Akaike Information Criteria (AIC). The models considered include logit, log, and "logflip" ($\log(1 - x)$) link and transformation functions, up to first order in the transformed domain, resulting in monotonic calibration functions. See Appendix A for additional details.

We find that the parametric forms for the calibration curve and distribution of scores are well captured by these simple GLM and Beta models. Figure 5(a,b) shows the resulting fits, with parameters summarized in Appendix A. We observe significant skew in the score distribution which, as discussed

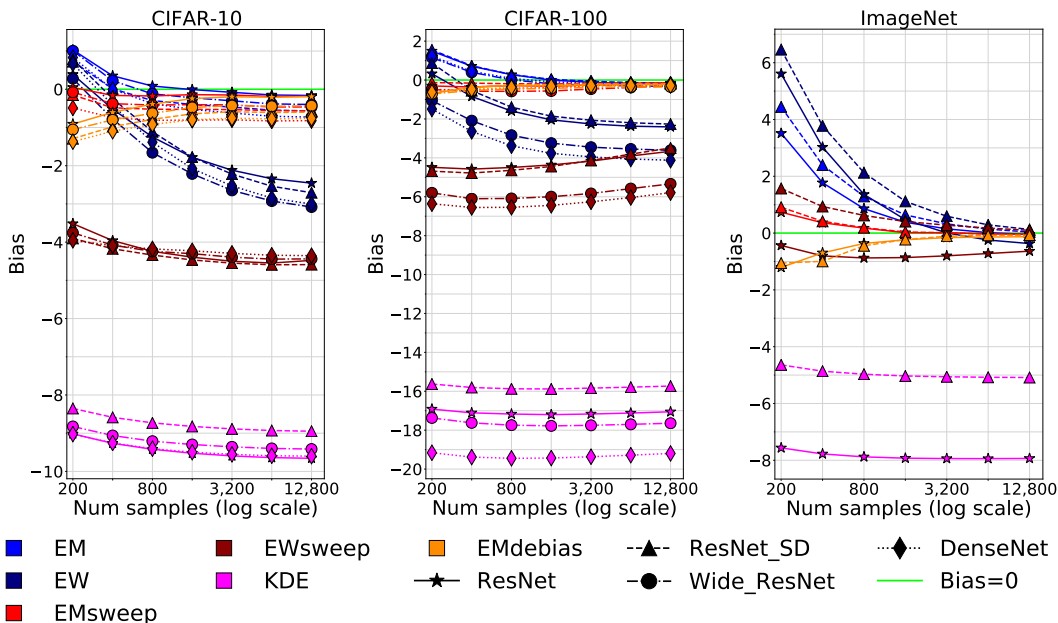

Figure 6: **EMsweep is less biased than alternative calibration metrics.** We plot bias versus number of samples $n$ for calibration metrics on simulated data drawn from the CIFAR-10, CIFAR-100, and ImageNet fits (Section 5.1). The dataset the model was trained on has a greater influence on bias than the model architecture. Metrics based on equal mass binning consistently outperform equal width binning. Exploiting monotonicity in the `EMsweep` metric helps the most at small sample sizes.

in Section 5.2, poses a challenge to measuring calibration error with equal-width bins. We find that the dataset has more influence on the fits than the neural model, with ImageNet models the least skewed and CIFAR-10 the most (correlating with model accuracy). Figure 4 5(c) indicates that $ECE_{BIN}$ scores computes on simulated data from the fits closely match $ECE_{BIN}$ scores computed on the real data, as witnessed by the fact that the points lie near the line of equality.

## 5.2 SIMULATION RESULTS ON DISTRIBUTIONS FIT TO CIFAR-10/100, AND IMAGENET

Using the resulting parametric fits from Section 5.1, we evaluate $ECE_{BIN}$ and $ECE_{SWEEP}$ using both equal mass binning and equal width binning and compare these values to the analytically computed TCE. In addition, we include a comparison to the recently proposed debiased estimator, $ECE_{DEBIAS}$, using equal mass binning (Kumar et al., 2019) and a smoothed Kernel Density Estimation (KDE) method for estimating calibration error (Zhang et al., 2020). We abbreviate each method as follows:

- `EW`: $ECE_{BIN}$ using equal width binning and 15 bins (Guo et al., 2017; Naeini et al., 2015),
- `EM`: $ECE_{BIN}$ using equal mass binning and 15 bins (Nixon et al., 2019),
- `EMdebias`: Debiased calibration metric using equal mass binning (Kumar et al., 2019)
- `KDE`: KDE calibration metric (Zhang et al., 2020)
- `EWsweep`: $ECE_{SWEEP}$ using equal width binning, and
- `EMsweep`: $ECE_{SWEEP}$ using equal mass binning.

We choose 15 bins for $ECE_{BIN}$ and $ECE_{DEBIAS}$, following the standard set by Guo et al. (2017). For a comparison of different choices of fixed number of bins, Appendix B includes an analysis of the bias and variance of $ECE_{BIN}$, $ECE_{SWEEP}$, and $ECE_{DEBIAS}$ across different bin numbers and sample sizes for the curves corresponding to CIFAR-10 ResNet-110, CIFAR-100 Wide ResNet-32 and ImageNet ResNet-152 models. We see that the optimal number of bins varies with the number of samples, and using a different fixed number of bins introduces bias at varying sample sizes.

Figure 6 plots the bias (estimated using $m = 1,000$ simulations) versus the sample size $n$ for the best-fit curves for each neural network model trained on the CIFAR-10, CIFAR-100, and ImageNet datasets. An unbiased estimator would have Bias = 0, which we highlight visually in green. We

find that the dataset the model was trained on has more influence on the calibration metric behavior than the model architecture, which may be unsurprising given that Section 5.1 shows that the dataset heavily influences both the distribution of confidence scores and the true calibration curve.

*Equal width versus equal mass binning:* Overall, metrics that employ equal mass binning show less bias than those with equal width binning. Surprisingly, for `EW` and `EWsweep` on CIFAR-10, as well as `EW` on CIFAR-100, we see increasing absolute bias as the number of samples increases across all model architectures tested. We propose a possible explanation for this phenomenon. As Figure 5 (a) shows, models trained on CIFAR-10 and CIFAR-100 have highly skewed confidence distributions and, as a result, equal width binning places the majority of the instances in the top bin. As we increase the number of samples, we increase the likelihood that we generate a sample that populates one of the lower bins, which, due to their low sample density, may have a poorer average estimate of the TCE.

*ECE_{SWEEP} versus alternative metrics:* Our experiments show that ECE_{SWEEP} with equal mass binning has either similar or less bias than alternative calibration error metrics, and at low sample sizes, the ECE_{SWEEP} method is consistently less biased than all other metrics. However, the ECE_{SWEEP} does not show improvements over other metrics when combined with equal width binning, and we do not recommend using the combination in practice.

*KDE estimator.* Compared to all calibration metrics we evaluate, the KDE estimator has much higher bias across the CIFAR-10, CIFAR-100, and ImageNet simulations. Our results suggest that the heuristic used to choose the kernel bandwidth and the specific 'triweight' kernel worked well for the one synthetic example evaluated in (Zhang et al., 2020), but fails to generalize to the more realistic synthetic examples we study. Specifically, Zhang et al. (2020) assumes a Gaussian distribution for $P(X|Y)$ and a logistic confidence score distribution, which result in notably different qualitative shapes than the logit distributions we obtain from models trained on CIFAR-10/100 or ImageNet (see Figure 5(a, b) or the reliability diagrams and confidence score distributions from Kängsepp (2019)).

*Debiased estimator.* The debiased estimator (Kumar et al., 2019) uses a jackknife technique to estimate the per-bin bias in the standard ECE_{BIN}, and subtracts off this bias to achieve a better binned estimate of the calibration error. However, unlike ECE_{SWEEP}, the debiased estimator still has a hyperparameter $b$ that controls the number of bins. On the CIFAR-10/100, and ImageNet simulations in Figure 6, the debiased estimator with 15 bins is more competitive to equal mass ECE_{SWEEP} than any other estimation method we test, but the equal mass ECE_{SWEEP} method still outperforms the debiased estimator for low sample sizes. In Appendix B, we also show that the equal mass debiased estimator has higher variance than the equal mass ECE_{SWEEP} (except when $b \leq 4$, when all estimators have high bias).

### 5.3 BIAS VERSUS TRUE CALIBRATION ERROR

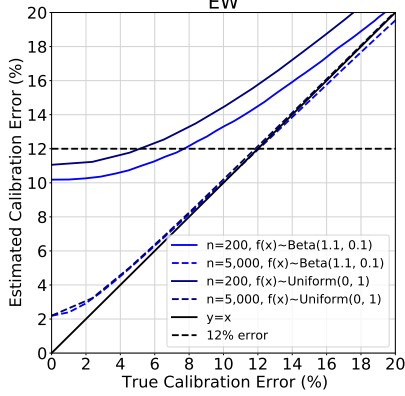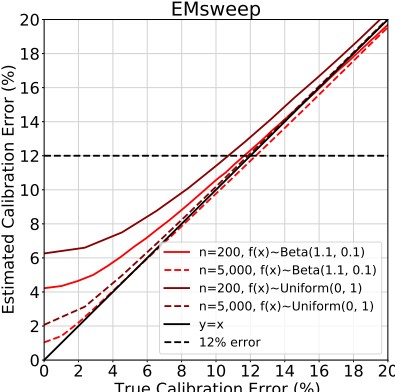

Figure 7: **Bias in calibration estimation increases as TCE decreases.** We plot average ECE (%) for `EW` (left) and `EMsweep` (right) versus the TCE (%), with varying sample size and score distributions. The estimator bias is systematically worse for better calibrated models, and the effect is more egregious with fewer samples. At $n = 200$ samples, depending on the score distribution, an `EW` estimate of 12% could either correspond to 5% or 8% TCE. The `EMsweep` metric is able to mitigate the bias and ambiguity in calibration error estimation to a certain extent.

We next evaluate the estimation bias of the baseline EW and our best estimator from the previous section, EMsweep, as we systematically vary the TCE. We are interested in the low calibration error regime because a goal of many recalibration algorithms is to reduce the calibration error of the model to 0%. Figure 7 shows the average estimated calibration error for EW and EMsweep versus the TCE. The average calibration error is computed across $m = 1,000$ simulated datasets, and we include results for two sample sizes, $n = 200$ and $n = 5,000$, and two score distributions, $f(x) \sim \text{Uniform}(0, 1)$ and $f(x) \sim \text{Beta}(1.1, 0.1)$, the beta distribution fit to the CIFAR-100 Wide ResNet_32. To control the TCE, we assume $\mathbb{E}_Y[Y \mid f(X) = c] = c^d$ and vary $d \in [1, 10]$. When $d = 1$ the true calibration curve is $\mathbb{E}_Y[Y \mid f(X) = c] = c$, which means the model's predicted confidence score is exactly equal to its empirical accuracy and thus the TCE is 0%. As we increase $d$, we move the true calibration curve farther away from the perfect calibration curve, which increases the TCE of the model.

The bias in the calibration error estimation can be seen visually as the difference between the ECE and the TCE. Perfect estimation (0 bias) corresponds to the $y = x$ line. Bias is highest when the model is perfectly calibrated (TCE is 0%) and generally decreases as TCE increases. Using a larger sample size of $n = 5,000$ reduces the bias, but when the model is perfectly calibrated, the $\text{ECE}_{\text{BIN}}$ can still be off by 2%. The EMsweep metric significantly reduces this bias.

In practice, we do not know the distribution of scores $\mathcal{F}$, the true calibration curve $\mathbb{E}_Y[Y \mid f(X) = c]$, or the TCE. So, given a measurement of calibration error, how much bias can we expect? If we measure an $\text{ECE}_{\text{BIN}}$ of 20%, we expect it to be fairly close to the TCE. However, if we see an $\text{ECE}_{\text{BIN}}$ of 2%, it is possible that the model may actually be perfectly calibrated! We further explore this dilemma in the next section.

### 5.4 How well can we detect miscalibration?

Consider the situation where we have a model whose TCE is unknown and we wish to test the hypothesis that the model is miscalibrated, i.e., TCE > 0. Our ability to detect miscalibration depends on the TCE, the sample size ($n$), and the method for estimating calibration error. We conduct a simulation with $f(x) \sim \text{Beta}(1, 1)$ and true calibration curve from the family $\mathbb{E}_Y[Y \mid f(X) = c] = c^d$, where $d$ is varied to obtain a range of TCE. Allowing for a type I error rate of .05 (also known as the false-alarm rate, or the rate of mistakenly claiming miscalibration when a model is perfectly calibrated), we obtain type II error rates (also known as the miss rate, or the rate of failing to detect a miscalibration). Figure 8 shows the type II error rate as a function of TCE and $n$ for EMsweep and EW. Our results indicate that EMsweep obtains a significantly lower failure rate than EW, particularly for under 10,000 samples. More generally, we note limitations with both methods: to detect a miscalibration of 2%, over 10,000 samples are needed; and if one has under 500 samples, the miscalibration must be greater than 10% to be detected reliably.

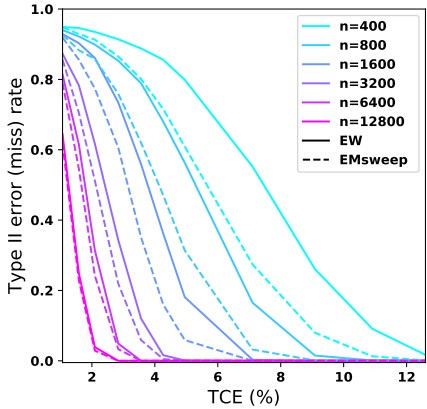

Figure 8: **Probability of failing to detect miscalibration (type II error, or miss rate),** plotted as a function of TCE for various sample sizes ($n$), with type I error rate fixed at 0.05. EMsweep (dashed lines) obtains a significantly lower failure rate than EW (solid lines).

## 6 Discussion

Much research in model calibration has focused on recalibrating models, i.e., transforming $f(x)$ to $f'(x)$ (Platt et al., 1999; Zadrozny & Elkan, 2001; 2002). We focus on estimating calibration error, because without a good estimate of TCE, there is little point in trying to recalibrate models. What implications do our results have on choosing and evaluating recalibration algorithms?

One possibility is that bias affects all recalibration methods in the same way, which would imply that we should still be able to select recalibration method A over method B with a biased estimator. However, our results show that the distribution of confidence scores significantly impacts the bias in calibration error estimation, even when the sample size is fixed. Since recalibration methods are inherently designed to modify the confidence score distribution, we cannot assume that bias will affect all methods in the same way.

Another possibility is that the bias is small compared to the calibration error differences we would measure. However, our results suggest that this is not true. Even when the number of evaluation samples is high, $n = 5{,}000$, Figure 7 shows that it is entirely possible that we might measure an $\text{ECE}_{\text{BIN}}$ of 2% when the model is in fact perfectly calibrated. Moreover, Figure 2 shows that the preference of recalibration algorithm can change depending on whether $\text{ECE}_{\text{SWEEP}}$ or $\text{ECE}_{\text{BIN}}$ is used to measure calibration error, implying that bias might meaningfully affect the conclusions of previous studies of calibration error such as those in Guo et al. (2017).

Several authors attempt a different approach to recalibration: improving model calibration during training. For instance, Mukhoti et al. (2020) trains a model with a batch size of 128 across multiple types of losses including maximum mean calibration error (Kumar et al., 2018) and Brier loss (Brier, 1950) which explicitly tries to minimize a calibration loss using 128 examples at a time. However, our results suggest that training a model with naive estimates of calibration error as an objective using a batch size $< O(1000)$ is a potentially flawed endeavor, particularly because the distribution of scores from the model is changing throughout training, and any potential measure of calibration may be more affected by the distribution of scores (as opposed to the calibration curve).

## 7    RELATED WORK

**Sensitivity of $\text{ECE}_{\text{BIN}}$ to hyperparameters.** Several works have pointed out that $\text{ECE}_{\text{BIN}}$ is sensitive to implementation details. Kumar et al. (2019) show that $\text{ECE}_{\text{BIN}}$ increases with number of bins while Nixon et al. (2019) find that $\text{ECE}_{\text{BIN}}$ scores are sensitive to several hyperparameters, including $\ell_p$ norm, number of bins, and binning technique. In addition, Nixon et al. (2019) find that $\text{ECE}_{\text{BIN}}$ with equal mass binning produces more stable rankings of recalibration algorithms, which is consistent with our conclusion that equal mass $\text{ECE}_{\text{BIN}}$ is a less biased estimator of TCE. However, in contrast to prior work, we study the sensitivity of the *bias* in $\text{ECE}_{\text{BIN}}$ to implementation hyperparameters.

**Metrics for calibration error estimation.** Gupta et al. (2020) propose a calibration error metric inspired by the Kolmogorov-Smirnov (KS) statistical test that estimates the maximum difference between the cumulative probability distributions $P(f(X))$ and $P(Y \mid f(X))$. The KS is similar to the maximum calibration error (MCE) (Naeini et al., 2015) in that it computes a worst-case deviation between confidence and accuracy, but the KS is computed on the CDF, while the MCE uses binning and is computed on the PDF. In contrast, our work focuses on measuring the *average* difference between confidence and accuracy. As mentioned in Guo et al. (2017), both the worst case and average difference are useful measures but may be applicable under different circumstances.

## 8    CONCLUSIONS AND FUTURE WORK

If we are to rely on the predictions from machine learning models in high stakes situations like autonomous vehicles, content moderation, and medicine, we must be able to detect when these predictions are likely to be incorrect. Given that the default confidence scores produced by machine learning models do not necessarily correspond to the model's empirical accuracy, recalibration is necessary in order to produce reliable and consistent output. However, it is impossible for a recalibration algorithm to achieve perfect calibration if we cannot measure calibration accurately. Our results show that the statistical bias in current calibration error estimators grows as we approach perfect calibration, but this bias can be mitigated by using monotonic estimation techniques. We conclude with some directions for future work:

**Simulation as an evaluation tool.** We have shown that simulation is a powerful technique for evaluating calibration error estimators. However, especially for small sample sizes, the $\text{ECE}_{\text{SWEEP}}$ method does not completely eliminate estimation bias, and we only evaluated a finite set of distributions arising from image classification datasets and models. We hope future work can use simulation as an effective tool for developing both new calibration metrics and new recalibration techniques, and that evaluations can be extended to a more diverse set of datasets and models.

**Distribution shifts.** Since models deployed in real world application will necessarily make predictions on out-of distribution examples and since we would like these predictions to be calibrated (Ovadia et al., 2019), it is important that we are able to accurately measure and improve calibration error on non-iid data. However, in such situations, we do not necessarily expect the calibration curve to be monotonic. Thus, exploring how distribution shifts can modify the shape of the true calibration curve is an important future direction.

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

## A    MAXIMUM-LIKELIHOOD FITS

Table 1 provides parameters fit to the top-scores obtained for each of 10 empirical datasets by maximum likelihood estimation.

Table 1: Parameters of best fit for distribution functions investigated in Section 5.1.

|                    | $\hat{\alpha}$ | $\hat{\beta}$ |
|--------------------|--------|--------|
| resnet110_c10      | 2.7752 | 0.0478 |
| resnet110_SD_c10   | 2.1714 | 0.0394 |
| resnet_wide32_c10  | 2.3806 | 0.0379 |
| densenet40_c10     | 1.9824 | 0.0397 |
| resnet110_c100     | 1.1823 | 0.1081 |
| resnet110_SD_c100  | 1.1233 | 0.1147 |
| resnet_wide32_c100 | 1.0611 | 0.0650 |
| densenet40_c100    | 1.0805 | 0.0808 |
| resnet152_imgnet   | 1.1359 | 0.2069 |
| densenet161_imgnet | 1.1928 | 0.2206 |

Global optimia $\hat{\alpha} \in [0, 200]$, $\hat{\beta} \in [0, 50]$ are approximately computed using a recursively-refining brute-force search until both parameters are established to within an absolute tolerance of $1e{-}5$. Each step in the recursion contracts a linear sampling grid ($N = 11$) by a factor of $\gamma = .5$ centered on the previously established optimal parameter, subject to the constraints $\alpha, \beta > 0$. Experiments confirmed that the computed optima were robust to the hyperparameters $N, \gamma$.

$$\arg\min_{\alpha,\beta} \sum_i -\ln \frac{x_i^{\alpha-1}(1-x_i)^{\beta-1}}{\frac{\Gamma(\alpha)\Gamma(\beta)}{\Gamma(\alpha+\beta)}} \tag{8}$$

Table 2 provides parameters fit to calibration functions. For each sample image $x_i$ in the image dataset, define $s_i = f(x_i)$ to be the score (the output of the top-scoring logit after softmax) and $y_i \in \{0, 1\}$ to be the classification ($y_i = 1$ when the top-scoring logit correctly classified image $x_i$) for the sample image. The loss for the binary generalized linear model (GLM) across different combinations of link functions $g(y)$ and transform functions $t(s)$ was optimized via the standard loss (Gelman et al. (2004)):

$$\arg\min_{b_0,b_1} \sum_i -\ln p_i^{y_i}(1-p_i)^{1-y_i}, \quad p_i = g^{-1}(b_0 + b_1 t(s_i)) \tag{9}$$

For each dataset, the GLM of best fit was selected via the Akaike Information Criteria using the likelihood at the optimized parameter values.

Table 2: Parameters of best fit for calibrations functions investigated in Section 5.1.

| dataset_name | glm_name | AIC | $b_0$ | $b_1$ | dataset_name | glm_name | AIC | $b_0$ | $b_1$ |
|---|---|---|---|---|---|---|---|---|---|
| resnet110_c10 | logflip_logflip_b0_b1 | **2779.22** | -0.24 | 0.30 | resnet110_SD_c100 | logit_logit_b0_b1 | **7873.61** | -0.88 | 0.49 |
| | logit_logflip_b0_b1 | 2790.40 | -0.55 | -0.38 | | logflip_logflip_b0_b1 | 7878.19 | -0.09 | 0.35 |
| | logit_logflip_b1 | 2827.51 | | -0.31 | | logflip_logflip_b1 | 7932.28 | | 0.38 |
| | logit_logit_b0_b1 | 2840.70 | -0.38 | 0.36 | | logit_logflip_b0_b1 | 7944.61 | -1.04 | -0.52 |
| | logit_logit_b1 | 2900.02 | | 0.30 | | logit_logit_b1 | 8315.51 | | 0.32 |
| | logflip_logflip_b1 | 2932.09 | | 0.34 | | log_log_b0_b1 | 8437.82 | -0.11 | 2.18 |
| | log_log_b0_b1 | 3221.72 | -0.06 | 2.53 | | logit_logflip_b1 | 8510.36 | | -0.30 |
| | logit_logit_b0 | 3799.63 | 1.99 | | | log_log_b1 | 9988.07 | | 3.30 |
| | logflip_logflip_b0 | 3811.98 | -2.13 | | | logit_logit_b0 | 10803.27 | 0.80 | |
| | log_log_b0 | 3829.05 | -0.13 | | | log_log_b0 | 10810.90 | -0.37 | |
| | logit_logflip_b0 | 3868.40 | 1.95 | | | logit_logflip_b0 | 10823.15 | -1.16 | |
| | log_log_b1 | 4281.78 | | 4.75 | | logflip_logflip_b0 | 10834.48 | 0.78 | |
| resnet110_SD_c10 | logit_logflip_b0_b1 | **2498.98** | -0.27 | -0.35 | resnet_wide32_c100 | logflip_logflip_b0_b1 | **7183.93** | -0.13 | 0.21 |
| | logit_logit_b1 | 2502.52 | | 0.30 | | logit_logit_b0_b1 | 7219.14 | -0.98 | 0.33 |
| | logit_logflip_b1 | 2508.70 | | -0.30 | | logflip_logflip_b1 | 7233.51 | | 0.25 |
| | logit_logit_b0_b1 | 2538.41 | -0.26 | 0.33 | | logit_logflip_b0_b1 | 7297.00 | -1.06 | -0.34 |
| | logflip_logflip_b0_b1 | 2550.29 | -0.36 | 0.27 | | logit_logit_b1 | 7626.21 | | 0.19 |
| | logflip_logflip_b1 | 2572.85 | | 0.35 | | log_log_b0_b1 | 7650.97 | -0.24 | 2.51 |
| | log_log_b0_b1 | 2594.91 | -0.08 | 1.98 | | logit_logflip_b1 | 7795.28 | | -0.17 |
| | log_log_b0 | 3137.19 | -0.19 | | | logflip_logflip_b0 | 8977.39 | -0.98 | |
| | logflip_logflip_b0 | 3150.42 | -1.80 | | | logit_logflip_b0 | 8987.38 | 0.49 | |
| | logit_logit_b0 | 3175.58 | 1.58 | | | log_log_b0 | 9000.24 | -0.49 | |
| | logit_logflip_b0 | 3179.67 | 1.56 | | | logit_logit_b0 | 9009.51 | 0.49 | |
| | log_log_b1 | 3697.37 | | 3.77 | | log_log_b1 | 11911.51 | | 5.48 |
| resnet_wide32_c10 | logit_logit_b1 | **2483.34** | | 0.26 | densenet40_c100 | logit_logit_b0_b1 | **8158.28** | -0.97 | 0.34 |
| | logflip_logflip_b0_b1 | 2487.69 | -0.47 | 0.22 | | logflip_logflip_b0_b1 | 8229.43 | -0.12 | 0.22 |
| | logit_logit_b0_b1 | 2511.39 | -0.13 | 0.28 | | logit_logflip_b0_b1 | 8267.77 | -1.08 | -0.35 |
| | logit_logflip_b0_b1 | 2558.45 | -0.26 | -0.28 | | logflip_logflip_b1 | 8368.86 | | 0.25 |
| | logit_logflip_b1 | 2586.47 | | -0.25 | | logit_logit_b1 | 8783.50 | | 0.19 |
| | log_log_b0_b1 | 2647.03 | -0.12 | 1.87 | | log_log_b0_b1 | 8832.20 | -0.25 | 2.26 |
| | logflip_logflip_b1 | 2713.17 | | 0.30 | | logit_logflip_b1 | 8918.57 | | -0.18 |
| | log_log_b0 | 2981.24 | -0.21 | | | logit_logit_b0 | 10138.24 | 0.47 | |
| | logflip_logflip_b0 | 2983.05 | -1.70 | | | logit_logflip_b0 | 10182.61 | 0.45 | |
| | logit_logit_b0 | 2989.90 | 1.49 | | | logflip_logflip_b0 | 10242.15 | -0.94 | |
| | logit_logflip_b0 | 3055.55 | 1.45 | | | log_log_b0 | 10261.01 | -0.50 | |
| | log_log_b1 | 4582.09 | | 4.61 | | log_log_b1 | 13322.10 | | 5.25 |
| densenet40_c10 | logit_logflip_b1 | **2910.62** | | -0.26 | resnet152_imgnet | logflip_logflip_b0_b1 | **18729.85** | -0.12 | 0.58 |
| | logit_logit_b0_b1 | 2961.31 | -0.40 | 0.31 | | logit_logit_b0_b1 | 18783.22 | -0.29 | 0.65 |
| | logit_logflip_b0_b1 | 3000.23 | -0.38 | -0.31 | | log_log_b0_b1 | 18785.44 | -0.03 | 1.32 |
| | logflip_logflip_b0_b1 | 3001.78 | -0.31 | 0.24 | | logflip_logflip_b1 | 18872.14 | | 0.65 |
| | logit_logit_b1 | 3021.54 | | 0.25 | | logit_logit_b1 | 19074.37 | | 0.57 |
| | logflip_logflip_b1 | 3027.78 | | 0.31 | | logit_logflip_b0_b1 | 19095.40 | -0.82 | -0.79 |
| | log_log_b0_b1 | 3153.38 | -0.12 | 2.04 | | log_log_b1 | 19840.25 | | 1.53 |
| | log_log_b0 | 3531.22 | -0.22 | | | logit_logflip_b1 | 20062.10 | | -0.50 |
| | logflip_logflip_b0 | 3589.11 | -1.60 | | | logflip_logflip_b0 | 26935.09 | -1.41 | |
| | logit_logit_b0 | 3601.85 | 1.37 | | | log_log_b0 | 26968.50 | -0.28 | |
| | logit_logflip_b0 | 3679.95 | 1.30 | | | logit_logflip_b0 | 27012.77 | 1.12 | |
| | log_log_b1 | 4735.18 | | 4.27 | | logit_logit_b0 | 27084.11 | 1.11 | |
| resnet110_c100 | logflip_logflip_b0_b1 | **8181.97** | -0.11 | 0.28 | densenet161_imgnet | log_log_b0_b1 | **18202.41** | -0.03 | 1.27 |
| | logit_logit_b0_b1 | 8206.19 | -0.88 | 0.39 | | logit_logit_b0_b1 | 18460.70 | -0.25 | 0.68 |
| | logflip_logflip_b1 | 8301.28 | | 0.31 | | logflip_logflip_b1 | 18521.48 | | 0.67 |
| | logit_logflip_b0_b1 | 8371.53 | -1.01 | -0.40 | | logflip_logflip_b0_b1 | 18534.07 | -0.10 | 0.61 |
| | logit_logit_b1 | 8732.11 | | 0.25 | | logit_logit_b1 | 18822.25 | | 0.60 |
| | log_log_b0_b1 | 8918.21 | -0.16 | 2.35 | | logit_logflip_b0_b1 | 18913.25 | -0.77 | -0.80 |
| | logit_logflip_b1 | 8926.99 | | -0.23 | | log_log_b1 | 19493.85 | | 1.44 |
| | logit_logflip_b0 | 10903.83 | 0.74 | | | logit_logflip_b1 | 19562.58 | | -0.54 |
| | logit_logit_b0 | 10943.95 | 0.72 | | | logit_logflip_b0 | 26426.38 | 1.19 | |
| | logflip_logflip_b0 | 10964.91 | -1.12 | | | logflip_logflip_b0 | 26445.91 | -1.46 | |
| | log_log_b0 | 11002.20 | -0.40 | | | logit_logit_b0 | 26519.76 | 1.18 | |
| | log_log_b1 | 11850.89 | | 4.26 | | log_log_b0 | 26662.65 | -0.27 | |

Table 3: ECE reported in Figure 5(c).

|  | ECE$_2$ (%) | \<ECE$_2$\> (%, simulated) |
|---|---|---|
| resnet110_c10 | 6.67 | 8.42 |
| resnet110_SD_c10 | 6.54 | 8.79 |
| resnet_wide32_c10 | 6.09 | 8.44 |
| densenet40_c10 | 6.70 | 8.09 |
| resnet110_c100 | 20.26 | 18.87 |
| resnet110_SD_c100 | 17.44 | 15.78 |
| resnet_wide32_c100 | 20.40 | 17.53 |
| densenet40_c100 | 23.12 | 19.69 |
| resnet152_imgnet | 6.85 | 9.26 |
| densenet161_imgnet | 6.15 | 6.87 |

## B  BIAS AND VARIANCE IN CALIBRATION METRICS

### B.1  BIAS

We evaluate bias for various calibration metrics using both equal-width and equal-mass binning as we vary both the sample size $n$ and the number of bins $b$. These plots should be seen as an alternative visualization to 6 where we additionally compare to different choices for the fixed number of bins $b$. Since the ECE$_{\text{SWEEP}}$ metrics adaptively choose a different number of bins for each sample size, we display the bin number for this metric as $-1$.

We find that ECE$_{\text{BIN}}$ can overestimate the true calibration error and there exists an optimal number of bins that produces the least biased estimator that changes with the number of samples $n$. Additionally, equal mass binning generally results in a less biased metric than equal width binning.

**CIFAR-10 ResNet-110.** Figure 9 assume parametric curves for $p(f(x))$ and $\mathbb{E}_Y[Y \mid f(X) = c]$ that we obtain from maximum-likelihood fits to CIFAR-10 ResNet-110 model output.

**CIFAR-100 Wide ResNet-32.** Figure 10 assume parametric curves for $p(f(x))$ and $\mathbb{E}_Y[Y \mid f(X) = c]$ that we obtain from maximum-likelihood fits to CIFAR-100 Wide ResNet-32 model output.

**ImageNet ResNet-152.** Figure 11 assume parametric curves for $p(f(x))$ and $\mathbb{E}_Y[Y \mid f(X) = c]$ that we obtain from maximum-likelihood fits to ImageNet ResNet-152 model output.

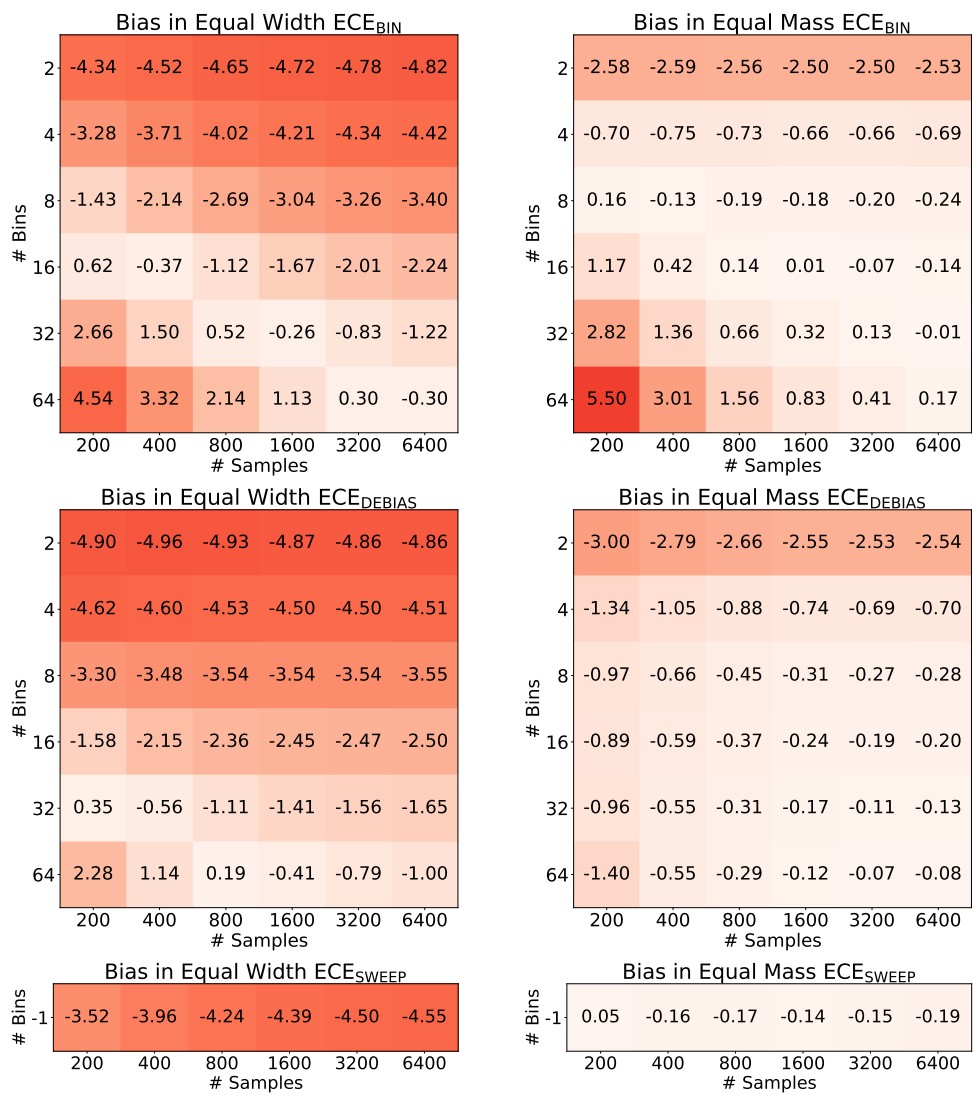

Figure 9: **Bias for various calibration metrics assuming curves fit to CIFAR-10 ResNet-110 output.** We plot bias for various calibration metrics using both equal-width binning (left column) and equal-mass binning (right column) as we vary both the sample size $n$ and the number of bins $b$.

## B.2  VARIANCE

We also compute the *variance* for various calibration metrics using both equal-width and equal-mass binning as we vary both the sample size $n$ and the number of bins $b$. As expected, the variance decreases with number of samples, but, unlike the bias, there is no clear dependence on the number of bins.

**CIFAR-10 ResNet-110.** Figure 12 assume parametric curves for $p(f(x))$ and $\mathbb{E}_Y[Y \mid f(X) = c]$ that we obtain from maximum-likelihood fits to CIFAR-10 ResNet-110 model output.

**CIFAR-100 Wide ResNet-32.** Figure 13 assume parametric curves for $p(f(x))$ and $\mathbb{E}_Y[Y \mid f(X) = c]$ that we obtain from maximum-likelihood fits to CIFAR-100 Wide ResNet-32 model output.

**ImageNet ResNet-152.** Figure 14 assume parametric curves for $p(f(x))$ and $\mathbb{E}_Y[Y \mid f(X) = c]$ that we obtain from maximum-likelihood fits to ImageNet ResNet-152 model output.

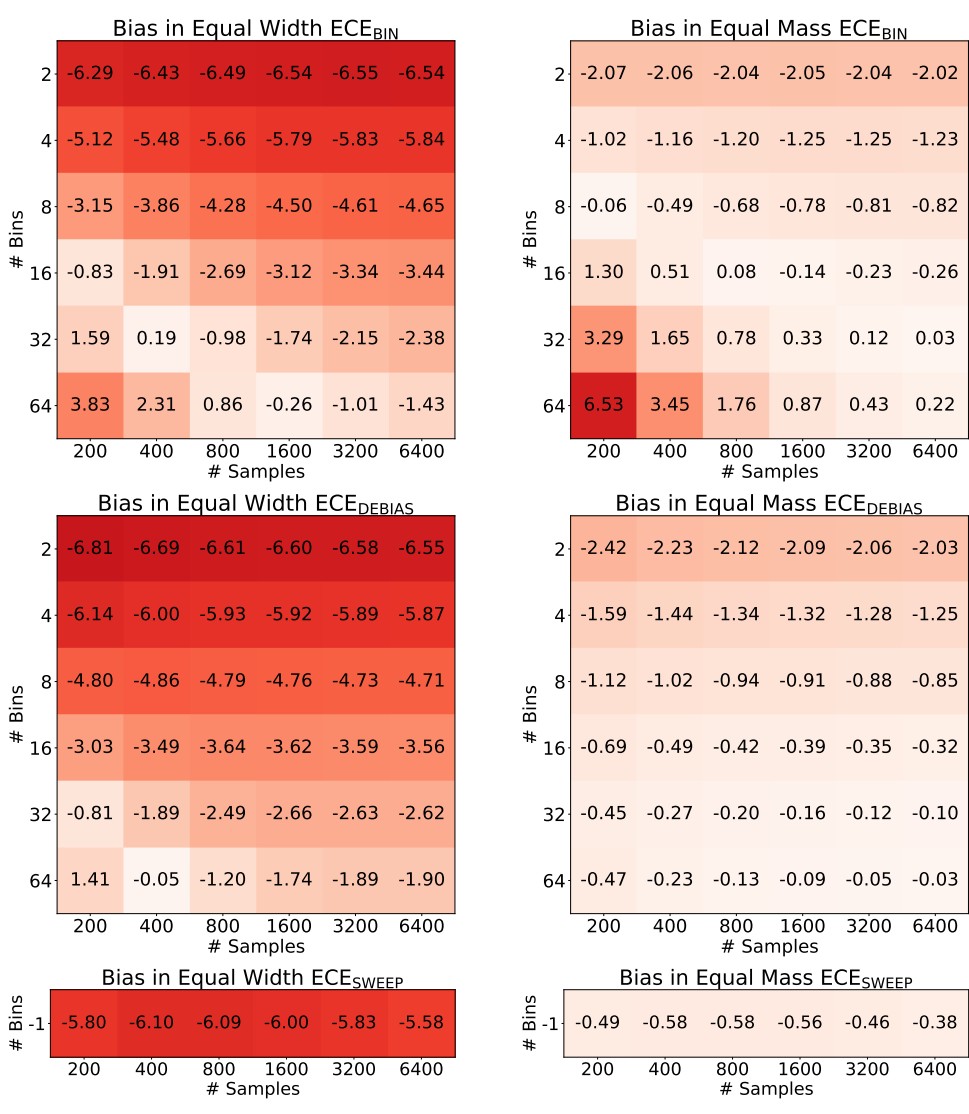

Figure 10: **Bias for various calibration metrics assuming curves fit to CIFAR-100 Wide ResNet-32 output.** We plot bias for various calibration metrics using both equal-width binning (left column) and equal-mass binning (right column) as we vary both the sample size $n$ and the number of bins $b$.

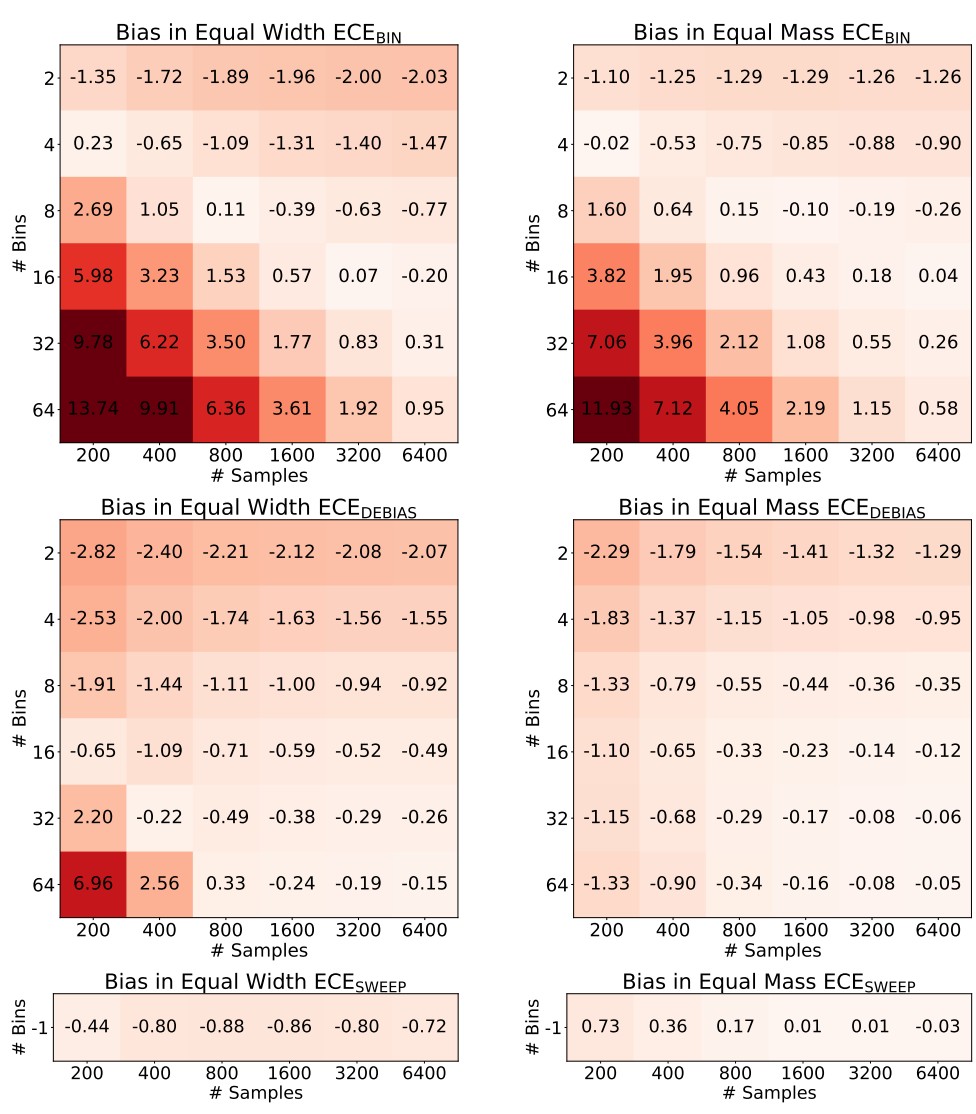

Figure 11: **Bias for various calibration metrics assuming curves fit to ImageNet ResNet-152 output.** We plot bias for various calibration metrics using both equal-width binning (left column) and equal-mass binning (right column) as we vary both the sample size $n$ and the number of bins $b$.

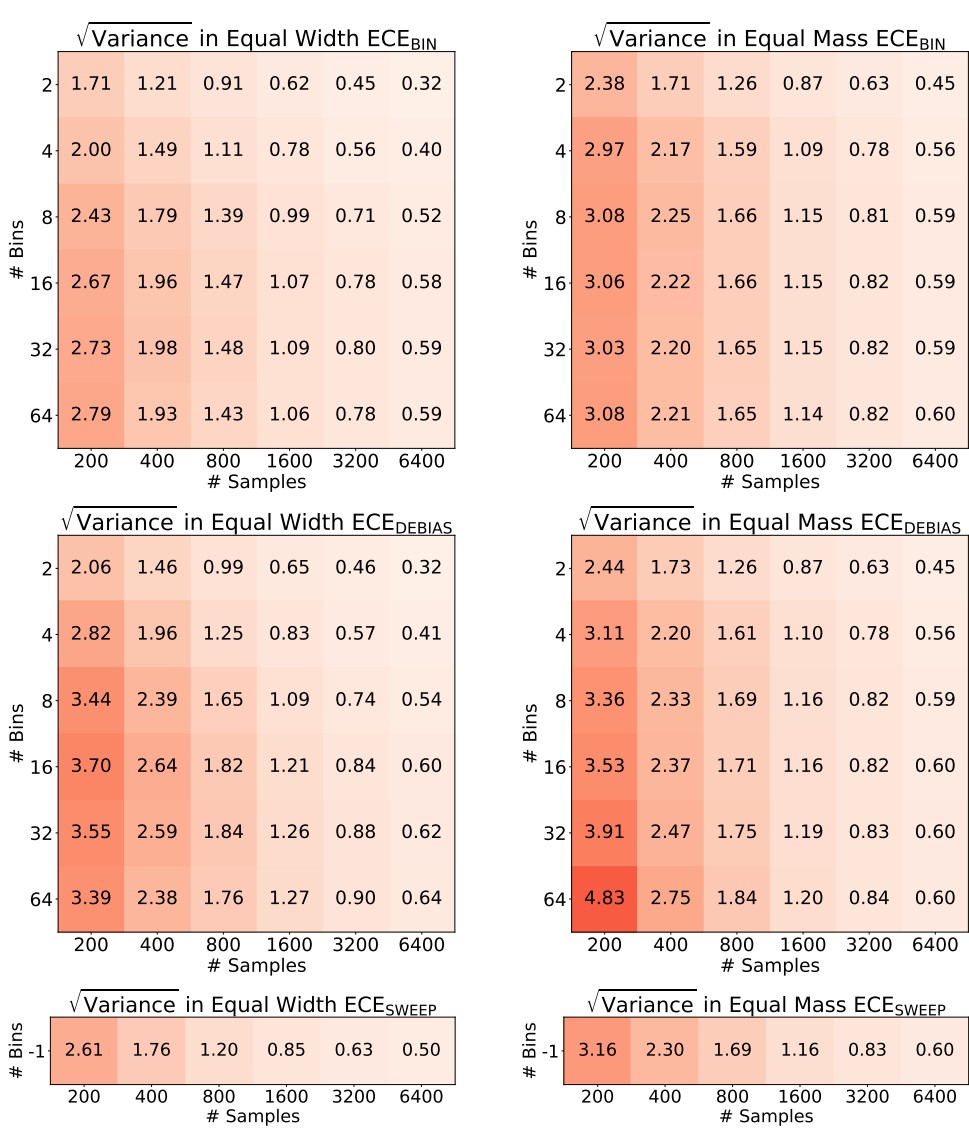

Figure 12: $\sqrt{\textbf{Variance}}$ **for various calibration metrics assuming curves fit to CIFAR-10 ResNet-110 output.** We plot $\sqrt{\text{Variance}}$ for various calibration metrics using both equal-width binning (left column) and equal-mass binning (right column) as we vary both the sample size $n$ and the number of bins $b$.

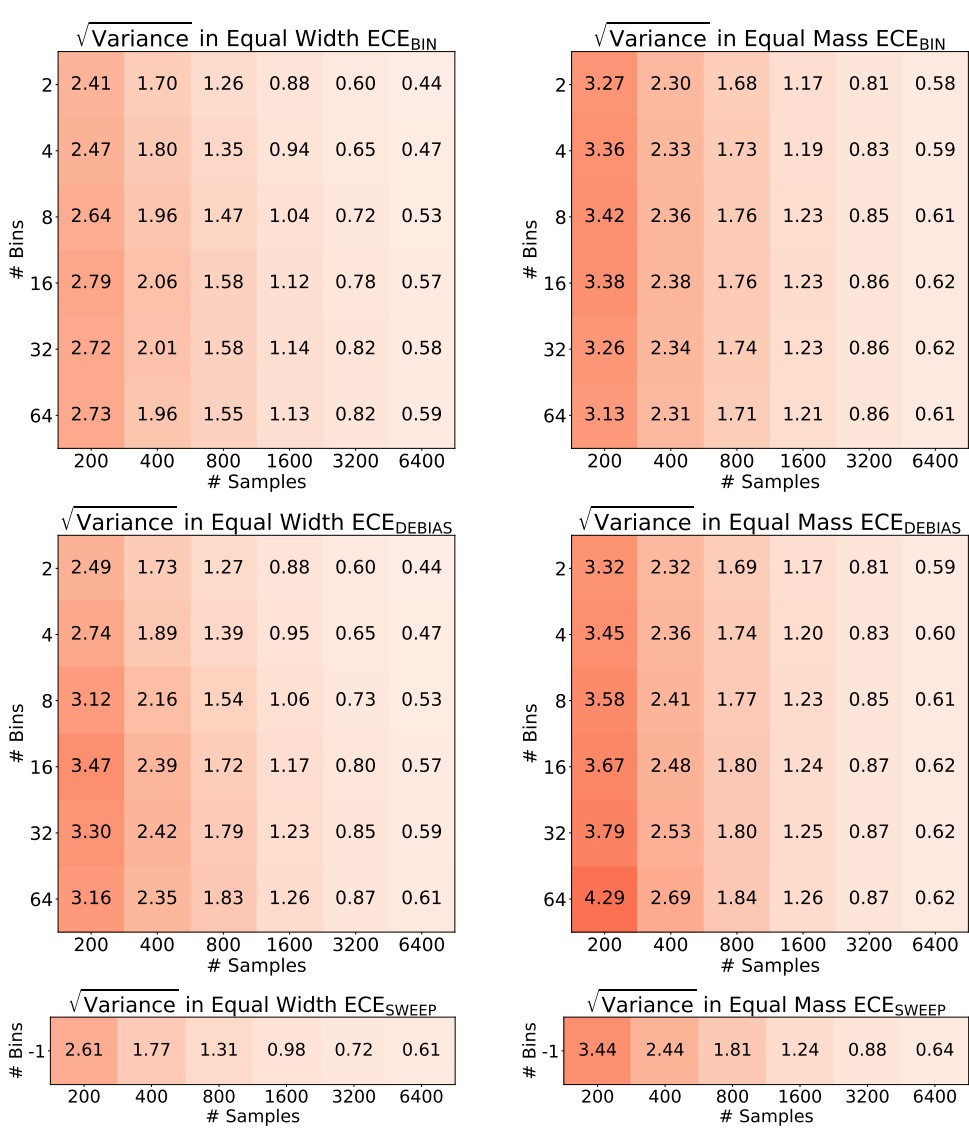

Figure 13: $\sqrt{\textbf{Variance}}$ **for various calibration metrics assuming curves fit to CIFAR-100 Wide ResNet-32 output.** We plot $\sqrt{\text{Variance}}$ for various calibration metrics using both equal-width binning (left column) and equal-mass binning (right column) as we vary both the sample size $n$ and the number of bins $b$.

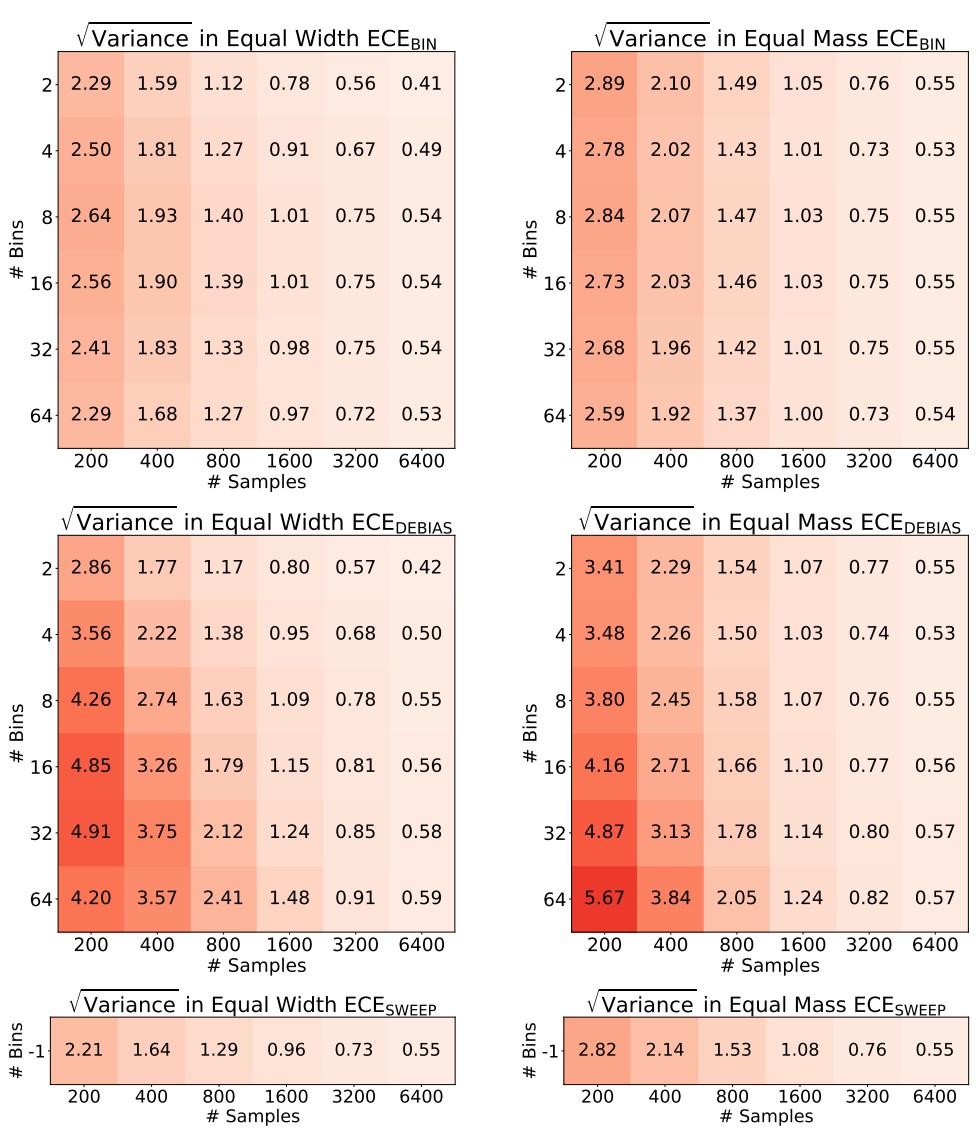

Figure 14: $\sqrt{\textbf{Variance}}$ **for various calibration metrics assuming curves fit to ImageNet ResNet-152 output.** We plot $\sqrt{\text{Variance}}$ for various calibration metrics using both equal-width binning (left column) and equal-mass binning (right column) as we vary both the sample size $n$ and the number of bins $b$.

## C   WHAT NUMBER OF BINS DOES EQUAL MASS $ECE_{SWEEP}$ CHOOSE?

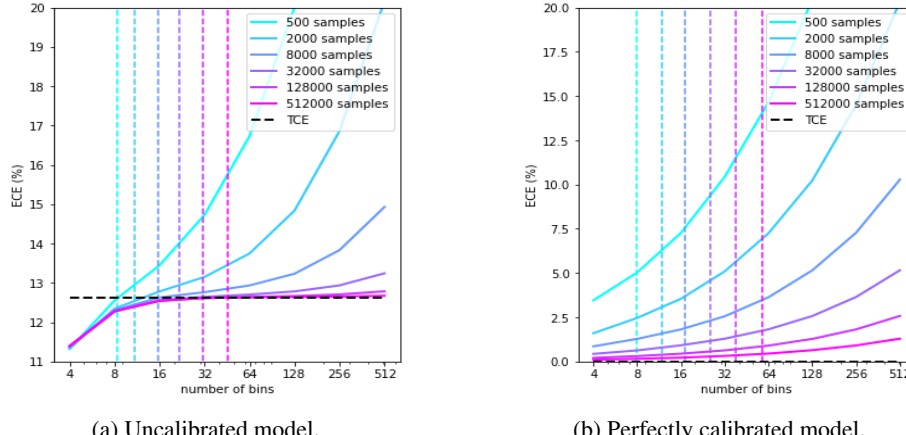

(a) Uncalibrated model.                    (b) Perfectly calibrated model.

Figure 15: **Bins chosen by equal mass $ECE_{SWEEP}$ method**. We plot equal mass $ECE_{BIN}$ % versus number of bins for various sample sizes $n$. We highlight the TCE with a horizontal dashed line and show the average number of bins chosen by the $ECE_{SWEEP}$ method for different sample sizes with vertical dashed lines. When the model is uncalibrated (left) $ECE_{SWEEP}$ chooses a bin number that is close to optimal. However, for perfectly calibrated models (right), the optimal number of bins is small (<=4), and $ECE_{SWEEP}$ does not do a good job of selecting a good bin number. The incorrect bin selection may partially explain why $ECE_{SWEEP}$ still has some bias for perfectly calibrated models. However, we note that any binning-based technique that always outputs a positive number will never be completely unbiased for perfectly calibrated models.

For Figure 15, the uncalibrated plot assumes $\mathbb{E}_Y[Y \mid f(X) = c] = \text{logistic}(10 * c - 5)$ while the calibrated plot assumes $\mathbb{E}_Y[Y \mid f(X) = c] = c$. Both experiments assume $f(x) \sim \text{Uniform}(0, 1)$.

# D   DIFFERENCES IN EW_ECE_BIN VS. EM_ECE_SWEEP

In Table 4 we compare the values for $ECE_{BIN}$ reported in Guo et al. (2017) Table 1 against our computation of the same quantities using 15 equal-width bins, but using the logits reported in Kängsepp (2019). We report both absolute and relative differences between these two quantities. The table has rows sorted according to the $ECE_{BIN}$ obtained in Guo et al. (2017).

Table 4: Comparison of ECE values (and associated rank orderings) computed using ew_ece_bin vs em_ece_sweep from uncalibrated logits.

| Uncalibrated ECE(%) | x=ew_bin | y=em_sweep | x-y | 100(x-y)/x |
|---|---|---|---|---|
| resnet110_SD_c10 | 4.11(0) | 4.10(0) | 0.01 | 0.24 |
| resnet_wide32_c10 | 4.51(1) | 4.48(1) | 0.03 | 0.66 |
| resnet110_c10 | 4.75(2) | 4.75(2) | -0.00 | -0.00 |
| densenet40_c10 | 5.50(3) | 5.49(3) | 0.01 | 0.13 |
| densenet161_imgnet | 5.72(4) | 5.72(4) | -0.00 | -0.00 |
| resnet152_imgnet | 6.54(5) | 6.54(5) | 0.00 | 0.00 |
| resnet110_SD_c100 | 15.86(6) | 15.83(6) | 0.03 | 0.18 |
| resnet110_c100 | 18.48(7) | 18.48(7) | 0.00 | 0.00 |
| resnet_wide32_c100 | 18.78(8) | 18.78(8) | -0.00 | -0.00 |
| densenet40_c100 | 21.16(9) | 21.16(9) | 0.00 | 0.00 |

Table 5: Comparison of ECE values (and associated rank orderings) computed using ew_ece_bin vs em_ece_sweep from logits calibrated using temperature scaling Guo et al. (2017). Red indicates differences in the sorted order of each entry in the column.

| Temp. scaling(%) | x=ew_bin | y=em_sweep | x-y | 100(x-y)/x |
|---|---|---|---|---|
| resnet110_SD_c10 | 0.56(0) | 0.36(1) | 0.19 | 34.99 |
| resnet_wide32_c10 | 0.78(1) | 0.21(0) | 0.57 | 72.65 |
| densenet40_c100 | 0.90(2) | 0.72(2) | 0.18 | 20.45 |
| densenet40_c10 | 0.95(3) | 0.90(3) | 0.05 | 4.98 |
| resnet110_c10 | 1.13(4) | 0.91(4) | 0.23 | 19.90 |
| resnet110_SD_c100 | 1.21(5) | 0.98(5) | 0.23 | 18.96 |
| resnet_wide32_c100 | 1.47(6) | 1.31(6) | 0.16 | 10.82 |
| densenet161_imgnet | 1.94(7) | 1.88(8) | 0.06 | 3.01 |
| resnet152_imgnet | 2.08(8) | 2.13(9) | -0.05 | -2.59 |
| resnet110_c100 | 2.38(9) | 1.87(7) | 0.51 | 21.38 |

Table 6: Comparison of ECE values (and associated rank orderings) computed using ew_ece_bin vs em_ece_sweep from logits calibrated using isotonic regression Zadrozny & Elkan (2002).

| Isotonic regression(%) | x=ew_bin | y=em_sweep | x-y | 100(x-y)/x |
|---|---|---|---|---|
| resnet110_SD_c10 | 1.03(0) | 0.70(0) | 0.33 | 32.15 |
| resnet_wide32_c10 | 1.19(1) | 0.77(1) | 0.42 | 35.41 |
| resnet110_c10 | 1.47(2) | 0.93(2) | 0.55 | 37.14 |
| densenet40_c10 | 1.68(3) | 1.61(3) | 0.08 | 4.66 |
| densenet161_imgnet | 4.64(4) | 4.64(4) | -0.00 | -0.00 |
| resnet110_SD_c100 | 4.89(5) | 4.86(5) | 0.04 | 0.81 |
| densenet40_c100 | 5.01(6) | 4.95(6) | 0.06 | 1.14 |
| resnet152_imgnet | 5.15(7) | 5.11(7) | 0.04 | 0.74 |
| resnet_wide32_c100 | 5.76(8) | 5.64(8) | 0.12 | 2.15 |
| resnet110_c100 | 6.19(9) | 6.05(9) | 0.15 | 2.40 |

Table 7: Comparison of ECE values (and associated rank orderings) computed using ew_ece_bin vs em_ece_sweep from logits calibrated using histogram binning Zadrozny & Elkan (2001). Red indicates differences in the sorted order of each entry in the column.

| Histogram binning(%) | x=ew_bin | y=em_sweep | x-y | 100(x-y)/x |
|---|---|---|---|---|
| resnet_wide32_c10 | 0.56(0) | 0.56(0) | 0.00 | 0.00 |
| resnet110_SD_c10 | 0.62(1) | 0.57(1) | 0.05 | 7.61 |
| resnet152_imgnet | 0.78(2) | 0.76(4) | 0.02 | 2.14 |
| densenet161_imgnet | 0.80(3) | 0.64(2) | 0.16 | 19.86 |
| densenet40_c100 | 0.81(4) | 0.80(5) | 0.01 | 1.02 |
| resnet110_c10 | 0.84(5) | 0.67(3) | 0.17 | 20.67 |
| resnet110_c100 | 0.91(6) | 0.91(6) | 0.00 | 0.00 |
| densenet40_c10 | 1.27(7) | 1.27(7) | -0.00 | -0.00 |
| resnet_wide32_c100 | 1.45(8) | 1.45(8) | 0.00 | 0.00 |
| resnet110_SD_c100 | 1.58(9) | 1.55(9) | 0.03 | 1.82 |

