# OpenReview forum: "Mitigating bias in calibration error estimation"
_ICLR.cc/2021/Conference — Reject_

### Official Review · AnonReviewer1 · 2020-10-14
**A fair point but hard to gauge the importance of the problem**

**Rating:** 4
**Confidence:** 4

**Review:**

In the manuscript, "Mitigating bias in calibration error estimation", the authors examine the problem of estimating the 'calibration error' of a binary classifier; the problem context being that addressed by techniques such as Platt scaling.  Here the class of estimators considered are what I would call 'bin smoothers', namely function estimators constructed from piecewise constant sub-estimators.  That these estimators suffer biases in the finite sample regime---and indeed asymptotically if the number of bins is not allowed to increase with sample size---does not seem remarkable to me (even without reference to the extensive literature on this class of problem), and likewise is noted (though not numerically explored) by previous authors considering the calibration error topic (e.g. Nixon et al. 2019).  The question then is whether this bias can be shown to have meaningfully affected the conclusions of previous studies of calibration error corrections (in particular, overturning or questioning the results of studies that were themselves important in the field).  From the experimental section presented here it is not clear that this is the case.  On the other hand, if the value of the 'sweep' method is to be the focus then a comparison should be made against a selection of competitive methods for bin smoothing in contemporary use (e.g. moving windows, adaptive bandwidths) rather than simply the equal width and equal mass bins of fixed number.

---

> ### Author Response · Authors · 2020-11-11
> **Request for references**
>
> Might the reviewer be able to point us to the relevant literature for the “competitive methods for bin smoothing in contemporary use (e.g. moving windows, adaptive bandwidths)"?

---

> ### Author Response · Authors · 2020-11-17
> **Response to R1's request for baseline comparisons**
>
> We thank the reviewer for their feedback. First, we address the reviewer's request to compare to methods for bin smoothing in contemporary use. From public comments on this paper, we learned of a method for bin smoothing that appeared in ICML 2020 which uses a KDE-based estimator (Zhang et. al. 2020). We updated Figure 4 in our paper to compare to the KDE estimator and another alternative debiased estimator proposed in Kumar et. al. 2019.  See general comment for discussion of results. We are unaware of other  bin smoothing calibration error estimation techniques that are in contemporary use.  We would welcome any additional suggestions that exist in the literature that we could compare against.
>
>  **Our experiments with moving-window-style estimator**. We also experimented with a “moving-window” style estimator for ECE that define a “soft neighborhood” around each example $f(x)$ , defined as $\mathcal{N}(f(x)) = \\{f(x’) :  ||f(x’) - f(x)||_2 <= \epsilon \\}$.  Here, epsilon is a hyperparameter that effectively controls the number of samples that are averaged together to form an estimate of the bin height.  However, it is not clear how to choose epsilon, and we concluded that this scheme just introduces a different hyperparameter (epsilon) analogous to the more traditional hyperparameter b. In simulations, we found that epsilon behaves just like the traditional hyperparameter b in that the bias varies systematically with epsilon and the setting that minimizes bias depends on the sample size and the dataset properties.
>
> **Comparison to other fixed choice of $b$**. We showed empirically in Figure 2 and 7 that the optimal number of bins varies with the number of samples, and using a fixed number of bins introduces bias at varying sample sizes. We have updated the paper to include a more thorough empirical analysis showing how bias and variance for ECE_bin and ECE_sweep varies  with number of bins and number of samples in Appendix B. Across varying bin numbers for ECE_bin, we show that ECE_sweep has lower bias than ECE_bin.

---

### Official Review · AnonReviewer4 · 2020-10-28
**Interesting sensitivity analysis but strong technical assumptions**

**Rating:** 4
**Confidence:** 4

**Review:**

The paper introduces an estimation approach for the true calibration error (TCE). TCE is compared to the binning-based estimator of calibration error (ECE) to measure bias.  Also, a maximum estimator of calibration error ECE_sweep is introduced and compared against standard binning ECE_bin across image datasets in classification tasks.

**Strengths**
-  With extensive sensitivity analysis, the paper illustrates the importance of  non-biased ${\rm ECE}$  calibration evaluation and model selection

**Weakness**

The reviewer appreciates the extensive sensitivity analysis. However, the contributions seem incremental in terms of methodology.

*Clarity*:
- The writing could be improved by the following:
1) Simulation steps for estimating bias in section 3  can be written in algorithm form
2) Computation or approximation of TCE described in section 5 is difficult to follow
3) Sections arranged in terms of concepts, e.g., TCE is introduced in section 2 and an approximation approach for TCE described in section 3 and section 5
4) The proposed ${\rm ECE}_{\rm sweep}$ estimator moved from the appendix to the main section
5) What is the motivation for the proposed ${\rm ECE}_{\rm sweep}$

*Strong technical assumptions*:
- Estimation of TCE relies on parametric formulations for confidence distribution and the true calibration curve;  such parametric assumptions are violated in practice
- The paper assumes monotonicity in the true calibration curve, which is ill-justified
- Provided that the proposed solutions rely on TCE, which is unknown. To evaluate bias, the proposed solution requires the above assumptions to hold

*Additional experiments*:
- How does the proposed solution compare to non-binning based approaches?
- Sensitivity analysis of top-r estimates, where r is the rank?

---

> ### Author Response · Authors · 2020-11-17
> **Response to R4**
>
> First, we thank the reviewer for their comments and suggestions for improvement. We respond to the weaknesses raised by the reviewer:
>
> **Contributions seem incremental in terms of methodology.** To summarize our paper’s main contributions, we believe that we are the first paper to identify and characterize systematic bias in popular existing calibration error estimation techniques. This bias has strong implications for any training objective that directly optimizes calibration. In particular, our work demonstrates how bias varies with number of bins and model output, and moreover is most severe for perfectly calibrated models. We also develop a technique to quantify bias using simulated datasets that reflect realistic model output.  Finally, we propose using monotonicity of the calibration curve to obtain better estimates of calibration error and contribute an alternative calibration metric, ECEsweep, which is simple to implement and has less bias than existing methods.
>
> *Clarity*:
> 1., 2. We will update the paper to include pseudo-code for estimating the bias in simulation and clarify how we approximated TCE in Section 5.
> 3. *Sections in terms of concepts.*  We chose to introduce the monotonic sweep method after the initial simulations in Section 3 because the need to select the bin number arises from our observations in Section 3 that bias varies systematically with sample size and number of bins
> 4. We moved the algorithm for ECE_sweep from the appendix to the main section.
> 5. *Motivation for ECE_sweep.*  In theory, the true calibration curve for models trained with cross-entropy loss is monotonic.  Empirically, in Figure 3 (b), we also verify that the true calibration curves of realistic model output is monotonic. ECE_sweep is able to be less biased than ECE_bin because it forces the binned estimate of the true calibration curve to be monotonic, which is an assumption that is both theoretically and empirically well grounded.  We will clarify this in the paper.
>
> *Strong technical assumptions:*
>
> -**Estimation of TCE relies on parametric formulations that are violated in practice.**  The metric we propose for practical use, ECE_sweep, does not rely on parametric assumptions for estimating the TCE.  Rather, we use these parametric assumptions only in our simulation test bed to evaluate the bias of calibration error estimators.  We visually confirmed that the parametric fits matched histograms of the raw data and we further verified our fits by visually confirming that the ECE numbers we could compute in simulation roughly matched the ECE numbers we computed on the real logits. We viewed these steps as necessarily for validating the parametric assumptions we made in order to estimate TCE.
>
> -**The paper assumes monotonicity in the true calibration curve, which is ill-justified.**
> If a model that is trained with standard loss functions like cross-entropy or logistic loss that maximize the likelihood ratio, it is theoretically well-established that such a model will have a monotonic calibration curve in the limit of inifinit data (\cite{green1966signal}, Section 2.3, chen2018calibration, gneiting2018receiver).  Standard algorithms for recalibration such as isotonic regression are based on the assumption that the true calibration curve is monotonic. Empirically, we also show in Section 5.1 that the model calibration curves  from several deep-learning architectures are well-modeled by monotonic fits.
>
> -**Provided that the proposed solutions rely on TCE, which is unknown. To evaluate bias, the proposed solution requires the above assumptions to hold.**
> We validated the parametric families that we used to compute TCE by visually comparing the resulting fits to logit histograms and by ensuring that the ECE we computed on the real logits was close to the ECE we computed on the simulated logits.
>
> *Additional experiments:*
>
> **How does the proposed solution compare to non-binning based approaches?**
> In the general comment, we describe our comparison to a non-binning based KDE estimator.

---

### Official Review · AnonReviewer3 · 2020-10-30
**Interesting ideas but baselines in experiments are deeply flawed.**

**Rating:** 7
**Confidence:** 3

**Review:**

Post Rebuttal:

Clearly I was too cavalier in suggesting alternative bin selection methods, as the authors have clarified. I think the paper is much improved by the comparison to other techniques such as ECE_debiased. The rebuttal also made me re-read with the view of seeing the demonstration of how bias varies with sample size as a major part of the contribution. In addition, the concrete demonstration of the relevance of the discussion via fig. 2 is nice.

On the whole, my opinion has shifted to be much more positive. I have been rating the paper somewhere between 6 and 7, and in such a case I would rather err on the side of acceptance.

Two asides:
1. I only realised this now, but I had misentered my confidence - it should have been 3 and not 5! I have corrected this now.

2. Of course, actual constants matter in practice, but one way to select number of bins by considering errors in $\overline{y}_k$ is that the squared error in each of these scales as $B/n,$ which, when added over $B$ bins induces error at scale $B^2/n$. With the same logic as the maximum number of bins in ECE_sweep, this suggests one way to set $B$ to be as $B = \lfloor \sqrt{n} \rfloor,$ which controls this error term. Of course, this is only a heuristic, and the analysis is incomplete because it's not accounting for correlations between these errors and $(f(x_i) - \overline{y}_k).$
In a larger sense, I still think that just looking at $15$ bins because that's all that has been looked at in the literature is not quite the right thing - basically I think that when studying an autotuning method, such as ECE_sweep, it is important to illustrate how it performs with respect to simple strategies based on some basic heuristics. This was the reason I wanted to score the paper at $6$, but I didn't bring this up when the authors asked for references and so felt it would be unfair to penalise on this basis.

-----
Summary of claims:

The paper studies the estimation of the calibration error of models, which is a problem of increasing relevance due to the rich settings that modern ML systems are being employed in. In particular, the focus is on mitigating the _bias_ in the commonly used binned estimators. The authors make the case that the biases of the binned estimators vary with the number of bins and the number of samples in a nontrivial manner, which is detrimental to point estimation. As a form of autotuning the number of bins that should be used, the  authors use the fact that the true calibration curve should be monotone increasing to propose choosing the number such that the resulting estimates of the average values of the $y$s in the resulting bins are monotonic. The bias of this estimator is then demonstrated on a number of simulations for which the laws of the classifier outputs and the calibration curves are obtained by fitting to models trained on standard datasets.

----

Strengths:

The problem being studied is certainly relevant, and the work is well contexualised. I found the idea of using the monotonicity of the $\{\\overline{y}_k\}$ as a criterion to choose an appropriate binning scale to be clever. I also like the problem of testing if the true calibration error is large or not. Finally, the method of section 5.1 is a nice way of generating realistic laws and calibration curves, and adds meaningful depth to the simulations.

----

Weaknesses:


1\) The nature of the contribution with respect to ECE_sweep is not clearly described in the text. Concretely, this amounts to a way to choose the number of bins using data (i.e., autotuning a hyperparameter in the estimate). While this, of course, leads to a different estimator, this is not something fundamentally different. I would much rather that the paper was upfront about the contribution. (In fact, I was pretty confused about the point the paper was making until I realised this).

2\) I don't think the baseline comparisons made in the experiments are appropriate. The proposal is a method to choose the appropriate number of bins in the estimate, and should be compared to other methods to do so instead of to an arbitrary choice of number of bins as is done in section 5.2. Without this comparison, I have no way to judge if this is a good autotuning method or not. Reasonable comparisons could be, e.g., choosing $b$ by cross validation, or, in equal mass binning, choosing $b$ so that each bin has a reasonable number of samples for the error $\overline{y}_k$ to not be too large.

3\) While the focus of the paper is on bias, it should be noted that by searching over many different bin sizes, the variance of  ECE_sweep may be inflated. If this is to such an extent that the gains in bias relative to other autotuning methods are washed out, then this estimator would not be good. To judge this requires at least that the variances for ECE_sweep are reported, but these are never mentioned in the main text.

4\) Choice of law in simulation in section 3, which are used to illustrate the dependence of bias on the number of bins, not aligned with the laws/curves in figure 3. Taking the latter as representative of the sort of laws and calibration curves that arise in practice, there are two issues:

4a\) The pdfs of $f$ tend to be a lot more peaked near the end than the one explored in section 3 - this is borne out by the values of $\alpha, \beta$ in the fits in Table 1. Beta(1.1,1) is remarkably flat compared to the curves in Fig 3.

4b\) There seem to be a few different qualitative properties of the calibration curves - monotone but with a large intercept at 0; those with an inflection point in the middle; and those with the bulk lying below the $y=x$ line. In particular, all of them tend to have at least some region above the $y=x$ line. The choice of curve $c^2$ in section 3 doesn't completely align with any of these cases, but even if we make the case that it aligns with the third type, this leaves two qualitative behaviours unexplored.

In fact, the choice of laws is such that the error of the hard classifier that thresholds $f$ at $1/2$ is $26\\%$. I don't think we're usually interested in the calibration of a predictor as poor as this in practice.

All of this makes me question the relevance of this simulation. Is the dependence of the bias on the number of bins as strong for the estimated laws as it is for these? Seeing the the equivalents of figs 7 and 8 for the laws from section 5 would go a long way in sorting this out.

5\) Experiments: As I previously mentioned, I don't think the correct baselines are compared to. Instead of posing the method against other autotuning schemes, just one choice of the number of bins is taken. This already makes it near impossible to judge the efficacy of this method.

Despite this, even the data presented does not make a clear case for ECE_sweep. In Fig. 4 we see that the bias of EW_sweep is even worse than EW. This already means that the sweep estimate doesn't fix the issues of ECE_bin in all contexts. It _is_ the case that EM_sweep has better bias than EM, but again, for samples large enough for the variances to be in control, it seems like these numbers are both converging to the same, so I don't see any distinct advantage when it comes to estimation. (of course, this is moot because this isn't the right comparison anyway)

Also, Fig. 5 is flawed because it compares EW and EM_sweep. It should either compare EM and EM_sweep, or EW and EW_sweep, I don't see why EW and EM_sweep are directly comparable.


Minor issues:

a\) Algorithm (1) and the formula for ECE_sweep in section 4 don't compute the same thing. In algorithm (1), you find the largest $b$ such that the resulting $\{\overline{y}_k\}$ is a monotone sequence, and return the ECE_bin for this number of bins. In the formula, you maximise the ECE_bin for all b that yield a monotone $\{\overline{y}_k\}$. From the preceding text, I assumed that the quantity in Algorithm (1) is intended.

b\) Why is the $L_p$ norm definition of the ECEs introduced at all? In the paper only $p = 2$ is used throughout. I feel like the $p$ just complicates things without adding much - even if you only present the $L_2$ definition, the fact that a generic $p$ can be used instead should be obvious to the audience.

c\) Design considerations for ECE_sweep - it is worth noting that accuracy is not all that we want in an estimate of calibration error. For instance, one might reasonably want to add this as a regulariser when training a model in order to obtain better calibrated solutions. One issue with ECE_sweep is that it introduces a problem in that how the number of bins in the ECE_sweep estimate changes with a small change in model parameters seems very difficult to handle, which makes this a nondifferentiable loss. Broader issues of this form, and a discussion of how they may be mitigated, could lead to a more well rounded paper.

----

Comments:

a\) Exact monotonicity in the ECE_sweep proposal - I find the argument stemming from the monotonicity of the true calibration curve, and the idea to use this to nail down a maximum binning size interesting. However, why should we demand exact monotonicity in the bin heights? Each $\\overline{y}_k$ will have noise at the scale of  roughly $\sqrt{b/n},$ (for equal mass binning with $b$ bins), and in my opinion, violation of monotonicity at this scale should not be penalised. Also, what if a few $\overline{y}_k$s decrease but most are increasing (i.e., the sequence has a few falling regions, but the bulk is increasing)? Perhaps instead of dealing with this crudely, the error of a shape constrained estimator may serve as a better proxy.

b\) Isn't the procedure for parametrically fitting the pdf of $f$, and $\\mathbb{E}[Y|f(X)],$ and then integrating the bias a completely different estimator for TCE of a model? In fact, if the laws are a good fit, as is claimed in section 5.1, then this plug in estimator might do well simply because the integration is exact. In fact, since the fit is parametric, this can further be automatically differentiated (if, say, $f$ were a DNN), and thus used to train.

c\) It would be interesting to see what number of bins are ultimately adopted in the ECE_sweep computations that are performed.

----

Overall opinion: The lack of comparison to appropriate baselines makes it near impossible for me to judge the validity of the proposed estimator. I feel like this is a deep methodological flaw when it comes to evaluating the main proposal of the paper. This is a real pity because I quite like some of the ideas in the paper.

Due to the inability to evaluate the main contribution of the paper, i am rating it a strong reject. I'd be completely open to re-rating it if appropriate comparisons are performed, and the case for the method is properly made.

---

> ### Author Response · Authors · 2020-11-11
> **Request for references**
>
> We are not aware of any literature that proposes tuning the number of bins-- might the reviewer provide some pointers?
>
> It is not currently clear to us how b can be chosen in cross validation since one doesn't necessarily know which ECE_BIN score resulting from a particularly b is least biased. Similarly, it is unclear how to estimate whether the error in the binned estimate is large in order to determine the appropriate number of samples needed in each bin.

---

> > ### Author Response · Authors · 2020-11-17
> > **Response to major issues**
> >
> > We appreciate the insightful and thorough feedback shared by this reviewer. We first respond to the weaknesses:
> > 1) **Nature of the contribution with respect to ECE_sweep is not clearly described.**
> > We agree with the reviewer that ECE_sweep is a technique for adaptively choosing the number of bins for the ECE_bin estimator.  It was not our intention to hide this fact, and we will change the writing to make this clearer earlier on in the text.
> >
> >    However, we would like to emphasize that ECE_sweep is only one contribution of the entire paper.  Most prominently, we believe that we are the first paper to identify and characterize systematic bias in popular existing calibration error estimation techniques. This bias has strong implications for any training objective that directly optimizes calibration. In particular, our work demonstrates how bias varies with number of bins and model output, and moreover is most severe for perfectly calibrated models. Finally, our work develops a technique to quantify bias using simulated datasets that reflect realistic model output.
> >
> > 2) **Automatically selecting number of bins.** As far as we are aware, there is no literature on automatic selection of number of bins; the need to select arises from our observation that bias varies systematically with sample size and number of bins.  We would welcome any additional suggestions that exist in the literature that we could compare against.
> >
> >    The reviewer proposes two example baselines: either selecting the number of bins b through cross-validation, or choosing $b$ so that each bin has a reasonable number of samples for the error $y_k$ to not be too large.  However, these baselines do not appear to have been proposed previously and, as we argue next, they do not have obvious implementations or advantages over ECE_sweep.
> >
> >    *Cross validation*. It is not apparent to us how to use cross-validation to select the number of bins that gives the least biased estimate of ECE_bin. As the simulation in Figure 2 shows, ECE_bin can either overestimate or underestimate the TCE depending on the number of samples and the number of bins. Thus, one cannot expect that choosing $b$ that yields the lowest ECE_bin on a held-out dataset will correspond to the $b$ that gives the least biased estimate of TCE.
> >
> >    *Techniques to control the error in the binned estimate*. Alternatively, the reviewer suggests we could choose $b$ so that each bin has a reasonable number of samples in order to control the error in each binned estimate.  We experimented with a similar style of estimator for ECE that define a “soft neighborhood” around each example $f(x)$ , defined as $\mathcal{N}(f(x)) = \\{f(x’) :  ||f(x’) - f(x)||_2 <= \epsilon \\}$.  Here, epsilon is a hyperparameter that effectively controls the number of samples that are averaged together to form an estimate of the bin height.  However, it is not clear how to choose epsilon, and we concluded that this scheme just introduces a different hyperparameter (epsilon) analogous to the more traditional hyperparameter $b$.
> > In simulations, we found that epsilon behaves just like the traditional hyperparameter b in that the bias varies systematically with epsilon and the setting that minimizes bias depends on the sample size and the dataset properties.
> >
> >    *Compare to more than one fixed choice of $b$*.  In comment (5), the reviewer also mentions we should compare to more than one choice of number of bins. We showed empirically in Figure 2 and 7 that the optimal number of bins varies with the number of samples, and using a fixed number of bins introduces bias at varying sample sizes. We have updated the paper to include a more thorough empirical analysis showing how bias and variance varies number of bins and number of samples for both the ECE_bin and ECE_debias method in Appendix B.

---

> > > ### Author Response · Authors · 2020-11-17
> > > **Response to major issues continued**
> > >
> > > 3) **Variances for ECE_sweep**.  As we mention in the text, we focused on the bias rather than the variance since the variance can be estimated from a finite set of samples through bootstrap resampling techniques whereas there is no corresponding procedure to estimate the bias. Moreover, empirically we found that the variance is relatively insensitive to the estimation technique and the number of bins.
> > > To thoroughly demonstrate these empirical findings, we have updated the paper to include full results in Appendix B.2 for the variance as a function of both the number of bins and the number of samples for the ECE_bin, ECE_sweep, and ECE_debias methods across simulations assuming curves corresponding to the CIFAR-10 ResNet-110, CIFAR-100 Wide ResNet-32, and ImageNet ResNet-152 model output.
> > >
> > >    Overall, we find that the ECE_sweep method has approximately 0.1% higher variance than the ECE_bin method (considering only bin numbers $b=8$ and $b=16$, since these bin numbers result in low bias for ECE_bin).   However, as our results in Figure 4 show, the reduction in bias from the equal mass ECE_sweep method is much more significant, especially at the low sample size of $n=200$.
> > >
> > >
> > > 4) **Choice of curves / law in simulation in section 3.**  The dependence of the bias on the number of bins is strong even under the more realistic laws.  We have updated the paper to include full results in Appendix B.1 for bias for varying sample size and number of bins for the ECE_bin, ECE_sweep, and ECE_debias methods across simulations assuming curves corresponding to the CIFAR-10 ResNet-110, CIFAR-100 Wide ResNet-32, and ImageNet ResNet-152 model output.
> > >
> > > 5) **Sweep estimate doesn’t fix the issue of ECE_BIN in all contexts; Fig. 5 is flawed because it compares EW and EM_sweep.** We compare against EW_ECE_BIN with 15 bins as the baseline since it is the current standard in the field and is commonly used for both model selection and ranking various recalibration methods (e.g. this is the metric used in Guo et. al.). Though equal-mass binning is not novel,  one of our contributions is to demonstrate that equal mass binning has significantly less bias (and only slightly more variance) when compared to equal width binning for both the ECE_BIN method and the ECE_SWEEP method.
> > >
> > >    We will include the corresponding results for EM_ECE_bin in Fig. 5 in the final version of the paper.

---

> ### Author Response · Authors · 2020-11-17
> **Response to minor issues**
>
> Minor issues:
>
> a) **Algorithm (1) and the formula for ECE_sweep in section 4 don't compute the same thing.**
> We intend that the ECE_sweep metric should compute the quantity in Algorithm 1.  We had conjectured that the algorithm and the formula were computing the same thing and that once a certain bin number resulted in a non-monotonic binning, increasing the bin number could not result in a monotonic binning. However, we now realize that this is not true and it is possible to design a counterexample. We thank the reviewer for catching this error and we corrected the mathematical formula for ECE_sweep to reflect Algorithm (1).
>
> b)**Why is the Lp norm definition of the ECEs introduced at all?**
> We include the Lp norm definition since p=1 is a popular choice for measuring calibration error using ECE_bin (p=1 is used in Naeini et. al, Guo et. al.) In addition, writing the true calibration error under the lp-norm matches the definition proposed in Kumar et. al. 2019.
>
>  c) **Design considerations for ECE_sweep. Can ECE_sweep be used as a differentiable loss?**
> One of our motivations for measuring bias in estimators at low sample sizes such as n=200 is to identify metrics that can be used to calibrate a model during training. (In this situation, a calibration loss would be computed over minibatches of examples.) We have developed a soft-binning variant of EM and EW (not described in our submission) that is differentiable. Each of these measures has a hyperparameter that could be set using the sweep method, although we have not yet investigated whether the sweep would be effective for a partially trained model.

---

> ### Author Response · Authors · 2020-11-17
> **Response to comments**
>
> a) **Relaxing exact monotonicity.** We did not experiment with relaxing exact monotonicity, although we did explore selecting the number of bins to be smaller or larger than the point of monotonicity violation. We agree with the reviewer that relaxing monotonicity holds promise, especially since we might not expect monotonicity to hold exactly for small sample sizes. We view this as a promising direction for future work.
>
> b)**Procedure for calculating TCE can be viewed as an estimate of calibration error.** While we agree with the reviewer, we did not recommend this technique for practitioners since verifying that the parametric assumptions are reasonable for the given data currently requires human supervision. For example, we visually confirmed that the parametric fits matched histograms of the raw data and we further verified our fits by visually confirming that the ECE numbers we could compute in simulation roughly matched the ECE numbers we computed on the real logits. Additionally, it is unclear whether the parametric families we assume for a particular sample will hold for the more general population.
>
> c) **What number of bins are ultimately adopted in the ECE_sweep computations?**
> In Appendix C, we include a visualization that shows the average number of bins chosen by the ECE_sweep method for varying number of samples.

---

### Official Review · AnonReviewer2 · 2020-10-31
**Improved Calibration Metric Based on Empirical Evidence**

**Rating:** 6
**Confidence:** 3

**Review:**

### Summary

This paper highlights a major flaw with the commonly-used $ECE_\text{BIN}$ calibration metric, namely that it is biased for perfectly-calibrated models. Through a large number of empirical experiments (including through simulation), the authors show that a newly proposed metric, $ECE_\text{SWEEP}$ is able to produce less biased estimates of calibration error.

### Paper Strengths

1. The paper has many empirical experimental and simulation data to support the claim that $ECE_\text{SWEEP}$ is a useful calibration metric.

2. The paper points drawbacks of commonly-used calibration metrics such as $ECE_\text{BIN}$, namely that it is biased for perfectly-calibrated models, and suggests alternatives.

### Major Concerns

1. In the theoretical discussion, the paper assumes binary classification. However, the empirical experiments are conducted on multi-class classification datasets. I did not find any mention of how the binary classification discussion is generalized to the multi-class setting.

2. The discussion about how the simulations were conducted was difficult to follow. Are the $m$ simulated datasets essentially subsets of the full CIFAR-10/100 or ImageNet datasets? (This seems to be implied, but is never made explicit. Otherwise, it can read as if these datasets are synthetic images.) And how do you draw samples such that "the confidence score distribution matches the neural model's best fit Beta distribution and the true calibration curve matches the neural model's best fit GLM"?

3. The equation for $ECE_\text{SWEEP}$ seems misleading. Are you really trying to maximize the quantity in parentheses over all possible values of $b$? And if so, why does that mean that this maximization will result in the largest number of bins, under the monotonicity constraint?

4. Is it always possible to satisfy the monotonicity constraint for $ECE_\text{SWEEP}$? Some more theoretically-sound discussion of $ECE_\text{SWEEP}$ would be much appreciated.

5. The discussion on Page 3 about Figure 2 can be expanded upon. Why does the "optimal bin count grow with the sample size"? This is stated as fact without explanation.

6. In general, while the experiments are mostly convincing, it would be useful to have some theoretical notions for why/when $ECE_\text{SWEEP}$ is less biased than $ECE_\text{BIN}$.


### Minor Concerns

1. There are many typos, grammatical errors, and mis-referenced figures throughout the manuscript. Please correct them.

2. Running experiments on new datasets is understandably time-consuming. However, given that this paper is heavily dependent on empirical evidence, it would be helpful to see experiments on datasets beyond image classification.

### Original Rating

**Rating** - 5: Marginally below acceptance threshold

**Confidence** - 3: The reviewer is fairly confident that the evaluation is correct

### Post-Rebuttal Update

I applaud the authors for providing detailed responses to my (and other reviewers') questions and for updating the manuscript appropriately. Several follow-up thoughts to my original questions:

1. **Binary classification.** I think I understand what you mean now. Basically you are still performing multi-class classification, but then you treat the calibration problem as a binary: is the top-1 model output correct or not? I think this still can be made clearer in the manuscript.

2. **Simulation procedure**: Thank you for the clarifications.

3. **$\text{ECE}_\text{SWEEP}$ equation**: I am still concerned that the equation given for $\text{ECE}_\text{SWEEP}$ differs from Algorithm 1. (Thanks for moving the algorithm into the main manuscript!) The issue is that I don't think Algorithm 1 is taking the maximum over the quantity in the parentheses of the $\text{ECE}_\text{SWEEP}$ equation. Algorithm 1 is not actually computing $\max_b g(b)$ for some function $g(b)$. Instead, it is first finding the maximum $b$ that satisfies some criterion, then calculating $g(b)$ at that selected $b$.

  If the equation and the algorithm are the same, it would imply that increasing the number of bins will increase the estimated calibration error. However, this does not seem to be true. Consider a binary dataset that is perfectly balanced: $\bar{y_1} = \frac{1}{n} \sum_{i=1}^n y_i = 0.5$, and consider a constant model, i.e. $\forall x: f(x) = 0.6$. Then for the case $b=1$, $ECE = 0.1$. But for the case $b=2$, $ECE = 0$ (assuming equal-width binning).

4. **Monotonicity constraint**: Thanks for pointing out the trivial constraint satisfaction.

5. **"optimal bin count grows with the sample size"**: I understand the empirical and intuitive reasoning for why this should be true. However, I still wish that this notion could be made more formal.

6. **Theoretical notions for why/when $\text{ECE}_\text{SWEEP}$ is less biased than other estimators of calibration error**: I am now more convinced that this is difficult to show, and I understand that a lot of the literature is based on empirical evidence. However, given that the results are empirical, I still would like to see experiments are non-image datasets.

**Updated Rating** - 6: Marginally above acceptance threshold

---

> ### Author Response · Authors · 2020-11-17
> **Response to R2**
>
> We thank the reviewer for the thoughtful comments and suggestions to improve our work.
>
> *Major concerns:*
>
> 1. **Generalization to multiclass setting.**  All empirical simulations in the paper assume binary classification.  When we fit parametric distributions to the realistic multiclass model output, we compute the top-label confidence score and whether or not the model’s predicted class corresponds to the true class, and we fit the GLM models and Beta distributions to this top-label model output. Then, in simulation, we create synthetic model output assuming a binary classification setup.
> Simulation procedure. When we create synthetic datasets, we model the $\textit{output}$ confidence scores of the model.  A single simulated dataset consists of $n$ model confidence scores $f(x) \in [0, 1]$ and the corresponding true labels for each example $y$.  This data is all that is necessary for computing calibration error.
>
>    To sample confidence scores that match the model’s best fit Beta distribution, we use numpy’s `np.random.beta(alpha, beta)`, setting alpha and beta to the parameters that we fit to the model’s top-label output via maximum likelihood estimation.   Then, to make sure the true calibration curve matches the neural model's best fit GLM, we first evaluate the model’s true calibration curve on the sampled score to compute the E[Y | f(X)=c] for that sample. Then, we generate a random label that has probability of being 1 corresponding to E[Y | f(X) = c]. We will update the paper to more clearly describe this procedure.
>
> 2. **Equation for ECEsweep is misleading.** We intend that the ECE_sweep metric should compute the quantity in Algorithm 1.  We had conjectured that the algorithm and the formula were computing the same thing and that once a certain bin number resulted in a non-monotonic binning, increasing the bin number could not result in a monotonic binning. However, we now realize that this is not true and it is possible to design a counterexample. We thank the reviewer for catching this error and we corrected the mathematical formula for ECE_sweep to reflect Algorithm (1).
>
> 3. **Monotonicity constraint.** Yes, it is always possible to satisfy the monotonicity constraint for ECEsweep since using one bin results in a monotonic binning. We updated the paper to include this clarification.
>
> 4. **Why does optimal bin count grow with sample size?** We observe empirically that the optimal bin count grows with the sample size in Figure 2.  The intuitive explanation is that having a large number of bins is generally preferred because we can obtain a finer-resolution estimate of the calibration curve. However, if we have a small number of samples, setting the number of bins too high may result in a poor estimate of the calibration curve due to the low number of samples in each bin.
>
> 5. **Theoretical notions for why/when ECESWEEP is less biased than ECEBIN.**  In theory, the true calibration curve for models trained with cross-entropy loss is monotonic.  Empirically, in Figure 3 (b), we also verify that the true calibration curves of realistic model output is monotonic. ECEsweep is able to be less biased than ECEbin because it forces the binned estimate of the true calibration curve to be monotonic, which is an assumption that is both theoretically and empirically well grounded.
>
> *Minor concerns*
>
> 1. We will carefully review the paper and correct typos and missing figure references for the final draft.
> 2. **Running experiments on datasets beyond image classification.** We agree with the reviewer and think this is a good suggestion.  For instance, it would be interesting to check the calibration curve and confidence score distribution for language or translation models, where the number of classes is particularly large.
>
>    We expect that as long as we are dealing with a classification task trained with cross-entropy loss, then, regardless of domain, the underlying true calibration curve of the model should be monotonic, implying that calibration estimators like ECEsweep which assume monotonicity should be less biased. The fact that recalibration algorithms which assume monotonicity (such as isotonic regression and Platt scaling) all work well for text domains (Guo et. al.) supports our hypothesis.

---

### Public Comment · ~Jize_Zhang1 · 2020-11-13
**Related work using KDE to mitigate the bias/binning issues in calibration error estimation**

Please check out our recent work on the use of KDE-based ECE estimator [1]. By replacing histogram with KDE, we provide a more reliable evaluation of the calibration error while mitigating the bias & binning sensitivity of existing histogram ECE estimators. The code is also available online.

[1] Jize Zhang, Bhavya Kailkhura, and T Han. "Mix-n-Match: Ensemble and compositional methods for uncertainty calibration in deep learning.", ICML 2020, https://arxiv.org/pdf/2003.07329.pdf

---

> ### Author Response · Authors · 2020-11-17
> **Comparison to KDE estimator**
>
> Thank you for sharing this reference. We were unaware of this work when we submitted the paper. We have now updated our paper and compare to the KDE estimator in Figure 4.

---

### Author Response · Authors · 2020-11-17
**Comparison to alternative baselines**

We thank all reviewers for the thoughtful comments and suggestions to improve our work. We first respond to the shared reviewer concern that our paper does not compare to appropriate baselines and we will respond individually to each reviewer addressing specific concerns.

Based on feedback,  we have updated Figure 4 in our paper and now compare to two additional baselines proposed in recent literature:

1. *KDE estimator from Zhang et. al. 2020*: Compared to all calibration metrics we evaluate, the KDE estimator has much higher bias across the CIFAR-10, CIFAR-100, and ImageNet simulations. Our results show that the heuristic used to choose the kernel bandwidth and the specific ‘triweight’ kernel worked well for the one synthetic example evaluated by the authors, but fails to generalize to the more realistic synthetic examples we study. Specifically, the authors assumes a Gaussian distribution for P(X | Y) and a logistic confidence score distribution, which result in notably different qualitative shapes than the logit distributions we obtained from models trained on CIFAR-10/100 or ImageNet (see Figure 3(a, b) in our paper or the reliability diagrams and confidence score distributions at https://github.com/markus93/NN_calibration/tree/master/reliability_diagrams).

2. *Debiased estimator from Kumar et. al. 2019*:  The debiased estimator uses a jackknife technique to estimate the per-bin bias in the standard ECEbin, and subtracts off this bias to achieve a better binned estimate of the calibration error.  However, the debiased estimator still has a hyperparameter that controls the number of bins $b$, and we show in Appendix B.1 for curves fit to realistic model output on CIFAR-10/100 and ImageNet that the bias is sensitive to the choice of $b$.

   On the CIFAR-10, CIFAR-100, and ImageNet simulations, we see that the debiased estimator with 15 bins is more competitive to equal mass ECEsweep than any other estimation method we test, but the equal mass ECEsweep method still outperforms the debiased estimator for low sample sizes. In Appendix B, we also include an analysis of the bias and variance of ECEbin, ECEsweep, ECEdebias methods across different bin numbers and sample sizes for curves corresponding to CIFAR-10 ResNet-110, CIFAR-100 Wide ResNet-32 and ImageNet ResNet-152 and find that the equal mass debiased estimator has higher variance than the equal mass ECEsweep (except when $b<=4$, when all estimators have high bias).

---

### Author Response · Authors · 2020-11-24
**Does the use of an improved ECE measure affect which recalibration method is preferred?**

Reviewer 1 asks whether the "bias can be shown to have meaningfully affected the conclusions of previous studies of calibration error corrections (in particular, overturning or questioning the results of studies that were themselves important in the field)".

We have updated the paper to include Figure 2, which examines this question using 10 pre-trained models, and compares the standard ECE measure, ECE_bin with 15 equal-width-spaced bins, to our ECE_sweep. With typical, large dataset sizes for recalibration and evaluation, we find that ECE_bin produces a different selection of the preferred recalibration
method on *30% of the models*. (We use histogram binning (Zadrozny & Elkan, 2001), temperature
scaling (Guo et al., 2017), and isotonic regression (Zadrozny & Elkan, 2002) as the recalibration
techniques.) When we reduce the size of the validation and evaluation by 10% and recalibrate with
these smaller sets, ECE_bin produces a different selection on 22% of the cases ( with 10 bins, we see
disagreement on 27% of the cases). Thus, the use of our improved ECE measure has significant
implications not only for estimation of calibration error but for improving calibration with methods
like temperature scaling.

We thank Reviewer 1 for helpful feedback that encouraged us to more thoroughly pursue this analysis.

---

### Decision · Program_Chairs · 2021-01-07
**Final Decision**

**Decision:**

Reject

**Comment:**

The paper proposes an adjustment to the ECE metric to make it less biased in the small sample case by including the assumption that the confidence output by a classifier is monotonic with the true correctness probability.  The main idea is to successively make finer bins until a non-monotonicity is observed.  The paper is interesting, but the magnitude of the contribution would be just enogh for a short paper if such a track existing in ICLR.

Reviewers have raised concerns about the discrepancy between their revised ECE formula and the Algorithm accompanying it, although that has been fixed through the author feedback phase.  Another concern is that for a paper whose core technical contribution is a revised metric for measuring calibration, a more thorogh empirical study over larger datasets is required.